# What Is ‘Muscle Health’? A Narrative Review and Conceptual Framework

**DOI:** 10.3390/jfmk10040367

**Published:** 2025-09-25

**Authors:** Katie L. Boncella, Dustin J. Oranchuk, Daniela Gonzalez-Rivera, Eric E. Sawyer, Dawn M. Magnusson, Michael O. Harris-Love

**Affiliations:** 1Muscle Morphology, Mechanics, and Performance Laboratory, School of Medicine, University of Colorado Anschutz Medical Campus, Aurora, CO 80045, USA; katie.boncella@cuanschutz.edu (K.L.B.);; 2Department of Physical Medicine and Rehabilitation, University of Colorado Anschutz Medical Campus, Aurora, CO 80045, USA; 3Eastern Colorado VA Geriatric Research, Education, and Clinical Center, Aurora, CO 80045, USA; 4Department of Bioengineering, University of Colorado Denver Anschutz Medical Campus, Aurora, CO 80045, USA; 5Preparation in Interdisciplinary Knowledge to Excel-Postbaccalaureate Research Education Program (PIKE-PREP), School of Medicine, University of Colorado Anschutz Medical Campus, Aurora, CO 80045, USA

**Keywords:** functional capacity, morphology, muscle performance, physiology, strength, tissue composition, muscle health

## Abstract

**Background**: Muscle health is an emerging concept linked to physical performance and functional independence. However, the term lacks a standardized definition and is often used as a broad muscle-related outcome descriptor. Clinical communication and research would benefit from a conceptual model of muscle health grounded in an established framework. **Methods**: We conducted systematic search and narrative synthesis to identify multifactorial measurement approaches explicitly described under ‘muscle health’. PubMed and CINAHL were searched for clinical and randomized controlled trials published in the past 5 years (final search: March 2025) that used the term “muscle health.” Studies were reviewed for explicit definitions of “muscle health,” and all identified outcomes (e.g., strength, mass) and measurement tools (e.g., grip strength, ultrasound) were synthesized. This review was retrospectively registered (INPLASY202580069). **Results**: Of the 65 clinical or randomized controlled trials that met inclusion criteria, 29 provided an operational definition of ‘muscle health’, while 36 inferred measurements without a clear definition. The identified measurements spanned four primary categories, with body composition/muscle mass being the most common (92.3%), followed by muscle performance (78.5%), physical function (63.1%), and tissue composition (30.8%). Most studies included more than one muscle health metric (93.9%). Common assessment methods included DXA (44.6%), grip strength (64.6%), and gait speed (27.7%). **Conclusions**: While there are common measurement approaches, the definition of muscle health varies widely in the cited works. The framework of the International Classification of Functioning, Disability and Health, was used to identify domains aligned with muscle health components of muscle morphology/morphometry (e.g., mass and composition), functional status (performance-based tasks), and physical capacity (muscle performance). This framework provides a structured basis for evaluating muscle health in research and clinical practice. Consistent use of these domains could enhance assessment and support efforts to standardize testing and interpretation across settings.

## 1. Introduction

Skeletal muscle tissue plays a critical role in maintaining overall health. Normal muscle function influences health in various ways, from regulating glucose and insulin homeostasis and storing amino acids to facilitating recovery from hospitalization and sustaining functional independence [1,2]. While the term ‘muscle health’ is widely used in research, it may denote various elements associated with muscle function that differ among researchers and practitioners [3,4]. For example, PubMed records show that use of the term ‘muscle health’ has increased substantially over the past two decades, with more than a fourfold rise in publications between 2010–2014 (5709 ± 1411 hits) and 2020–2024 (16040 ± 1130 hits) (Figure 1), with >14,000 hits through September 2025 alone, highlighting its growing prominence. However, while increasing at a substantial rate (like ‘reproductive health’, ‘joint health’, and ‘bone health’, but behind ‘cardiovascular health’ [see Figure 1]), muscle health remains inconsistently defined compared to more established constructs. Although components of muscle health are typically listed when the term is used in a study, standardized or operational definitions are rarely provided, or the term is used inconsistently [5]. Moreover, the frameworks used to provide theoretical constructs of muscle health are seldom provided in clinical studies [2,5]. Frameworks typically outline key constructs and their interrelationships, often drawing on existing literature, models, or theories. However, there is no consistent approach to the framework and components of muscle health when applied to clinical evaluations or outcome measurements in research settings. Without clear models or frameworks for muscle health, we will continue to observe a lack of proactive approaches to detect and manage common forms of muscle dysfunction associated with chronic disease and geriatric syndromes [5,6]. This lack of clarity is particularly concerning given the rapid rise in publications using the term muscle health (see Figure 1). As with more established constructs such as bone health and cardiovascular health, the growing use of the term without a standardized framework risks diluting its meaning, creating confusion in both clinical and research contexts. Establishing a definition and conceptual model of muscle health at this stage is therefore critical to ensure that its growing prominence is matched by scientific rigor and clinical utility.

The traditional geriatric vital signs obtained during a physical examination include blood pressure, pulse, respiratory rate, and temperature [7]. Nevertheless, others have proposed expanding the geriatric physical examination by including additional tests and screening measures related to cognition, walking speed, and strength assessment [6,8,9]. Furthermore, minimally time-consuming muscular performance tests (e.g., grip strength) may be warranted in the general population to serve as a proxy for longevity, quality of life, and cellular health [10,11,12]. The proposed expansion of the geriatric examination to include measures of muscle performance reflects the need to progress towards a standardized definition of muscle health. Ideally, establishing a standardized definition of muscle health precedes the attainment of consensus on key tests and measures as well as approaches to specific test protocols and data interpretation. Selected tests and measures must be aligned with components (e.g., categorical assessments of muscle tissue, muscle performance, and functional performance) that characterize accepted domains of muscle health. In turn, the domains associated with muscle health should be aligned with established conceptual frameworks regarding physical health and general principles that guide the physical examination process. The relationship among frameworks, domains, components, and assessment is depicted in Figure 2. Clarity regarding the framework for muscle health and approaches to objective measures that provide utility in both clinical and research settings would aid the clinical management of muscle dysfunction in a variety of patient populations. A viable framework requires an understanding of how skeletal muscle tissue impacts physical health and determining selected tests and measures that appropriately characterize muscle tissue and physical performance.

### 1.1. What Is Health?

The concept of ‘health’ now encompasses physical, mental, and social well-being, rather than solely the absence of disease, illness, and disability [13]. John Ware and colleagues [14] further expand upon this view by describing multiple health dimensions comprising two global health measures: mental and physical health. Physical health encompasses being free from diseases or ailments that result in physical impairments, performing daily activities and functional tasks without restriction, and having the capacity for physical activity through adequate strength, flexibility, and endurance [15]. Ware et al. [14] have further indicated that the dimensions of overall physical health include physical functioning and limitations due to physical challenges. Multiple investigators have observed that declines in muscle strength are frequently associated with diminished performance in activities of daily living (ADL), such as bathing, dressing, eating, toileting, and ambulation, and instrumental activities of daily living (IADL), which include more complex tasks including managing finances, shopping, meal preparation, housekeeping, and medication management [16,17].

This review broadly focuses on physical health, emphasizing how skeletal muscle tissue impacts physical functioning. Physical function (i.e., purposeful movement encompassing both basic and more complex tasks), requires complex interactions involving the musculoskeletal and nervous systems with support from the respiratory, cardiovascular, endocrine, skeletal, and integumentary systems [18]. Engaging in functional tasks and other forms of physical activity may demand the requisite muscle strength and endurance, but also dexterity, coordination, visual acuity, and balance. While functional assessments alone cannot confirm muscle impairments, functional assessments used in conjunction with other physiological measures can aid in the identification of various forms of muscle dysfunction.

Older adults tend to be most impacted by muscle dysfunction, with 35% percent of adults aged 65 years or above not being able to complete at least one ADL, and 53% not being able to complete at least one IADL [19]. In addition, estimates of low muscle mass-typically defined using appendicular lean mass indexed to height squared, with European Working Group on Sarcopenia in Older People (EWGSOP) recommended cut-points of <7.0 kg/m^2^ in men and <5.5 kg/m^2^ in women [20], and poor muscle composition have significant positive associations with poorer ADLs and IADL performance in older adults [21,22,23,24,25,26]. The emerging efforts to describe and assess muscle health specifically examine the role of skeletal muscle as a facilitator or inhibitor of physical health and the performance of functional tasks. Therefore, the assessment of muscle health should include direct or surrogate measures of skeletal muscle tissue that may range from morphology and morphometry to estimates of muscle mass. Identifying muscle pathology, poor muscle composition, or low muscle mass may aid the differential diagnosis process in clinical settings and identify when skeletal muscle significantly contributes to diminished physical health [2,6,24].

### 1.2. In Search of a Muscle Health Assessment Framework

Muscle health may be viewed as a subset of physical health. Given the interrelationship of these health concepts, the framework suggested by Koipysheva et al. [27] for assessing physical health pertains to muscle health. This assessment approach includes: (1) a physical examination, which may comprise anthropometric and/or physiologic measures (e.g., body composition estimates and/or muscle tissue morphological assessments), and (2) tests of “motor qualities” that are associated with functional tasks and physical capacity (e.g., functional tests and muscle performance measures). The application of this framework to assess muscle health is consistent with established typologies classifying health and related domains, such as the International Classification of Functioning, Disability and Health (ICF) [18,28], which delineates components of health and selected health-related aspects of well-being (Figure 1). Domains of the ICF include 1) ‘Body Functions and Structures’, and (2) ‘Activities and Participation’ [18]. Considering muscle health within the context of the ICF, using the assessment approach suggested by Koipysheva and colleagues [27], ‘Body Functions’ may be represented by measures of muscle performance; ‘Body Structures’ include estimates of muscle mass and/or muscle composition; and ‘Activities’ can be assessed through observed tests of physical performance using functional tasks. The ICF framework was selected because of its broad adoption in clinical and rehabilitation contexts, international recognition, biopsychosocial framework, and provision of a standardized taxonomy that facilitates comparison across populations and health conditions. In contrast, earlier conceptual models, such as the Nagi model, which primarily emphasized progression from pathology to disability [29], did not offer the same level of operational detail for the non-linear integration of body-level impairments, tissue characteristics, and activity limitations. Additionally, the ICF is a progression from the original International Classification of Impairments, Disabilities, and Handicaps model and has greater adoption than other disablement models such as the Institute of Medicine Enablement-Disablement Process Model.

Consequently, the objectives of this narrative review were to determine how researchers define and evaluate muscle health in contemporary literature and to determine if the outcome measures in the cited works in this review align with the proposed muscle health framework. Our goal was to gather data to support consensus efforts regarding a common framework and standardized approach to defining muscle health. Establishing a standardized approach to assessing muscle health could enhance the identification of muscle dysfunction, support proactive strategies to address the consequences of muscle aging, and facilitate the use of common methodologies within this area of study.

## 2. Materials and Methods

A narrative review was conducted by identifying papers to better understand the conceptual and operational definition of ‘muscle health’ used by other investigators and document the assessment tools used. These study data and definitional terms were then extracted and combined where appropriate to synthesize the current use of the term ‘muscle health’. This information was then interpreted using the ICF framework to develop a proposed conceptual model.

### 2.1. Reporting and Registration

The narrative review procedure and reporting were completed partially in line with the Preferred Reporting Items for Systematic Reviews and Meta-Analyses (PRISMA) [30]. The present review was retrospectively registered with the International Platform of Registered Systematic Review and Meta-analysis Protocols (INPLASY202580069; DOI: 10.37766/inplasy2025.8.0069).

### 2.2. Eligibility

Research studies must have included muscle health assessment as an element of clinical investigation. Articles that were excluded are non-human studies, case studies, review articles, or studies lacking outcomes that characterize muscle tissue and/or muscle performance. Otherwise, the conceptual nature of this review led us to include any randomized controlled or clinical trial, regardless of demographics (e.g., age, disease-state, athletes).

### 2.3. Information Sources and Search Strategy

Research articles were searched on the CINAHL and PubMed databases. The keyword ‘muscle health’ was searched. From those results, the articles were filtered only to include clinical or randomized controlled trials completed in the last five years (initial search from May 2023; final search to March 2025, with the full text available in English. Database results were downloaded and transferred to the Zotero reference manager (v6.0; Corporation for Digital Scholarship, Vienna, VA, USA). Covidence (v2627; Covidence, Melbourne, Australia; https://www.covidence.org/) was used to import all selected articles from the initial search. Duplicate articles were removed for appraisal. A total of three reviewers participated. Reviewers determined if the outcomes measured muscle health and how it was defined and measured in that study. Covidence was used to import and divide the literature among the reviewers. Every article imported was first screened independently based on the title and abstract by two reviewers. The criteria for the title/abstract screen were that (1) the article mentions ‘muscle health’ in the title/abstract, and (2) it is evident that the study featured outcome measures associated with muscle tissue and/or muscle function. We intentionally limited our search to studies that explicitly used the term ‘muscle health’, as the primary aim of this review was to examine how this emerging terminology is being defined and operationalized. A third reviewer was utilized if disagreements arose based on the eligibility criteria. Three independent reviewers then reviewed the full texts of the articles. The article must have provided an operational or conceptual definition of ‘muscle health’ as a criterion for full-text review. An additional reviewer was utilized if disagreements arose based on the eligibility criteria.

### 2.4. Data Extraction

The review of selected publications included the study premise, the population being studied, whether an operational or conceptual definition of ‘muscle health’ was provided, and how muscle health was being measured. The study’s outcome was also included in the summary tables (Table 1) if an operational definition was provided. Study characteristics were entered and analyzed in an Excel spreadsheet (Microsoft Corporation, Redmond, WA, USA). Muscle health definitions were stripped of non-essential words (e.g., ‘and’, ‘the’, ‘along with’, ‘characterized by’), with continuous terms connected with dashes (e.g., ‘muscle mass’ vs. ‘muscle-mass’). Key terms from each operational definition were categorized into five general ‘muscle health’ components: ‘body composition’, ‘physical function’, ‘muscle performance’, ‘tissue composition’, and ‘other’. A word cloud visualization with component-based color coding was generated using OpenAI’s ChatGPT (version-4o, April 2025) to script and render the figure in Python (version 3.11), utilizing the WordCloud (version 1.9.x) and Matplotlib (version 3.8.x) libraries.

### 2.5. Data Analysis and Interpretation

We employed a mixed-methods synthesis approach, combining quantitative analysis of measurement domains with structured qualitative description, to develop a proposed model of muscle health using the ICF framework. Each identified study was reviewed for the inclusion of assessments of body composition, tissue composition, muscle performance, and functional performance. Common language from operational definitions was extracted and analyzed in an Excel spreadsheet (see Figure 4). Absolute frequencies of inclusion across these domains were calculated, with relative (percentages) reported in-text. These data were interpreted in conjunction with the ICF framework and prior theoretical models [27], enabling us to identify recurring elements and their contextual applications. While we did not conduct a formal qualitative synthesis (e.g., thematic or framework analysis), the frequency, co-occurrence, and descriptive integration of domains across studies informed the proposed components (body/muscle tissue composition, muscle performance, functional performance) for the model. The development of the conceptual muscle health model employed a flexible approach that provides proposed domains and component categories suitable for both clinical and research applications. Nevertheless, the final selection of tests and measures used to assess muscle health components and the recommended data interpretation standards are beyond the scope of this work. The identification of assessment standards consistent with a conceptual model of muscle health is subject to further research and future consensus efforts. A risk-of-bias analysis was unnecessary due to the conceptual/narrative nature of the present review.

## 3. Results

### 3.1. Search Results

The search strategy and results are summarized in Figure 3. The original search (up to May 2023) resulted in 261 studies gathered between databases CINAHL (*n* = 198) and PubMed (*n* = 63). Thirteen studies were removed due to duplicate studies between databases. After the initial title and abstract screen, 158 studies were excluded. A full-text review was performed of the remaining 87 studies, with 43 studies included in the review [31,32,33,34,35,36,37,38,39,40,41,42,43,44,45,46,47,48,49,50,51,52,53,54,55,56,57,58,59,60,61,62,63,64,65,66,67,68,69,70,71,72,73]. Thirty-nine were excluded because the studies failed to meet the criteria for measuring or defining muscle health. Three studies were excluded due to insufficient study design: one due to an unfinished study and two due to access issues. The final search (May 2023 to March 2025) resulted in 90 hits (CINAHL = 73; PubMed = 17), with 11 duplicate pairs. Following title and abstract screening, 22 studies were accepted and added to the original batch of included studies [74,75,76,77,78,79,80,81,82,83,84,85,86,87,88,89,90,91,92,93,94,95]. In total, 65 studies were included in this review [31,32,33,34,35,36,37,38,39,40,41,42,43,44,45,46,47,48,49,50,51,52,53,54,55,56,57,58,59,60,61,62,63,64,65,66,67,68,69,70,71,72,73,74,75,76,77,78,79,80,81,82,83,84,85,86,87,88,89,90,91,92,93,94,95].

### 3.2. Identified Study Characteristics

A total of 16,249 participants were included (*n* = 7534 males, *n* = 8628 females), with 55 studies including both males and females [31,32,33,34,36,39,40,42,43,44,46,47,48,50,51,52,53,54,55,56,57,58,59,60,61,62,63,64,65,66,67,68,70,71,72,74,75,76,77,78,79,80,81,82,83,84,85,86,88,89,91,92,93,94,95], and four [35,41,45,69], and six [37,38,49,73,87,90] studies being exclusively male and female, respectively. The included studies investigated a wide range of populations, spanning children as young as four years old [45] to adults in their 90s [48,79,92], with an average reported age of 61.6 ± 16 years. However, the majority (*n* = 43, 61.5%) of studies recruited older participants (≥60 years) [32,33,34,37,38,39,40,42,43,47,48,51,52,53,54,55,56,57,59,62,64,65,66,67,72,73,75,78,79,80,81,83,84,85,86,87,89,90,91,92,93,94,95], while the participants of 18 studies (27.7%) ranged from 36 to 59 years [31,35,36,41,44,46,49,50,58,60,61,63,69,70,71,74,76,82], and another four studies (6.1%) had participants with a mean age ≤ 35 years [45,68,77,88]. Most studies (*n* = 33, 50.8%) focused on apparently healthy or community-dwelling individuals [32,33,34,35,37,39,41,44,47,49,50,51,53,55,57,58,59,61,64,66,67,72,73,75,77,80,81,86,90,91,92,93,94], while clinical cohorts (*n* = 24, 37%) encompassed diverse conditions including cancer, COPD, CKD, musculoskeletal and neurological disorders, or chronic illness [31,38,40,42,43,45,48,52,54,56,62,63,65,68,69,70,71,74,76,78,84,85,89,95]. Only one study examined athletic (golfers) participants [82]. Seven studies (10.8%) included mixed or unclear populations [36,46,60,79,83,87,88].

Of the 65 included studies [31,32,33,34,35,36,37,38,39,40,41,42,43,44,45,46,47,48,49,50,51,52,53,54,55,56,57,58,59,60,61,62,63,64,65,66,67,68,69,70,71,72,73,74,75,76,77,78,79,80,81,82,83,84,85,86,87,88,89,90,91,92,93,94,95], all measured one or more component of muscle health, which included the categories: body composition, muscle tissue composition, muscle performance and functional tasks. Twenty-nine studies provided an operational definition of ‘muscle health’ [31,35,36,39,41,43,45,49,50,53,55,56,59,62,63,64,65,69,70,71,73,75,76,78,83,86,87,88,89], while the other 36 assessed ‘muscle health’, but did not state an operational definition [32,33,34,37,38,40,42,44,46,47,48,51,52,54,57,58,60,61,66,67,68,72,74,77,79,80,81,82,84,85,90,91,92,93,94,95]. Studies with an operational definition are summarized in Table 1, while those lacking an operational definition are summarized in Table 2.

**Table 1 jfmk-10-00367-t001:** Summary of studies with a definition of ‘muscle health’ included.

Study	Population	Defined	Measured	Body Composition	Tissue Composition	Performance	Functional Tasks	Other
Anderson et al., 2022 [31]	Patients undergoing surgery for lumbar spine pathology, *N* = 54 (32 M:22 F), 51.5 ± 16.9 yrs	“Muscle health and function is influenced by structural features such as size (cross-sectional area) and tissue composition (e.g., amount of fatty infiltration within the muscle compartment)”, “…paraspinal muscle health (size and composition)…”	Muscle size, composition, and gene expression	CSA, mCSA, F-CSA, the proportion of fat within the muscle compartment (MRI)	Muscle, adipose, loose collagen, and dense collagen composition (tissue biopsy)			42 genes associated with adipogenic/metabolic, atrophic, fibrogenic, inflammatory, and myogenic pathways, 40S Ribosomal Protein (RPS18) and Beta-Actin (ACTB) as controls
Bathgate et al., 2018 [35]	One pair of male monozygotic twins, 52 yrs	“Skeletal muscle health—Whole muscle size, strength, and power were assessed. Additionally, protein and gene expression were measured for various markers of fiber type, metabolism, growth, repair, and inflammation.”	Skeletal muscle size, composition, strength, and power, molecular markers of muscle health, cardiorespiratory and pulmonary health, and blood profiles	CSA, MT (B-mode US) Lean mass, FM, total body fat percentage, visceral adipose tissue, bone mineral content, and bone mineral density (DXA)	Echo intensity (US) Muscle fiber composition—MHC isoforms, MyMHC expression, cellular metabolism (muscle biopsy, VL) Skeletal muscle fiber type, oxidative metabolism, citrate synthase, angiogenesis, vascular endothelial growth factor, muscular growth and repair, mechano-growth factor, insulin-like growth factor, myoblast determination protein 1, inflammatory responses (QRT-PCR)	Knee extension (dynamometry) and grip strength	Five sprints (Monark ergometer) and WAnT (Anaerobic capacity)	Cardiorespiratory: Resting heart rate, blood pressure, VO2max, and pulmonary function Muscle biopsy, AMPK protein expression Tracked normal physical activity patterns and dietary intake
Bauer et al., 2024 [75]	Community-dwelling older adults ≥70 yrs with urinary tract symptoms, *N* = 641 (264 M:377 F), 75.5 ± 4.4 yrs	“…age-related declines in skeletal muscle health, such as loss of muscle mass, volume, and strength/power, and related physical performance.”	Body size, muscle mass and volume, strength, power, physical function, cognition, and QoL	WBLM (D_3_-creatine) and thigh fat-free muscle volume (MRI)		Knee extension peak power (Keiser Air 420 exercise machine) and grip strength	400 m usual walking speed, SPPB, and four-square step test	Mobility Assessment Tool-short form, MoCA, CESD-10, EQ-5D, and CHAMPS Lower urinary tract symptoms: Lower Urinary Tract Dysfunction Research Network Symptom Index-10 Total energy expenditure, BMI
Berry et al., 2019 [36]	Adults with lower back pain, *N* = 14 (7 M:7 F), 52.8 ± 14.8 yrs	“The primary outcome measures of muscle health were mCSA and FF.”	mCSA, FF, strength, pain, and disability	mCSA (MRI)	FF (MRI)	Maximum lumbar extension strength (dynamometry)		Range of motion (isokinetic dynamometer), 100 mm visual analog scale, Oswestry Disability Index
Bhandari et al., 2025 [88]	Cancer survivors >2 years in remission and off therapy, *N* = 20 (10 M:10 F), 35 (18–67) yrs	“Exercise has been shown to improve muscle health, including muscle mass, strength, and function…”	Muscle mass, composition, strength, function, metabolic variables	Whole body fat and fat-free mass, segmental muscle mass, visceral adipose tissue (BIA)	RF, GM, GL: CSA, muscle thickness, IMAT (US)	Grip strength	SPPB	Blood: HbA1c, fasting glucose, HOMA-IR, myostatin BMI, waist circumference
Cegielski et al., 2022 [39]	Healthy adults, *N* = 37 (21 M:16 F), 72 ± 5 yrs	“…functional muscle health parameters (e.g., handgrip strength, leg strength, muscle mass by DXA imaging) …” “… established measures of muscle health (handgrip strength, 1-RM and MVC)…”	Muscle mass, strength, function, and metabolic variables	Thigh FFM (DXA)	Muscle thickness, fascicle length, and pennation angle (US)	Unilateral leg extension 1-RM, MVC (dynamometry) and grip strength	SPPB	Blood: MPS, MPB, and ASR (COSIAM) Muscle biopsy Urine sample collected to measure D_3_-creatinine
Davis et al., 2021 [41]	Men over a 15-year span, *N* = 522, 50.0 (IQR: 38.3–59.7) yrs	“Low muscle mass and poor muscle strength and function are key characteristics of poor muscle health.”	Muscle mass, strength, and function	SMI, whole body composition, and ALM (DXA)			TUG	Self-reported dietary data: food frequency questionnaire Self-reported physical activity: Baecke Physical Activity Questionnaire BMI
Distefano et al., 2024 [89]	Knee osteoarthritis patients, *N* = 655 (280 M:375 F), 76.1 ± 4.9 yrs	“Muscle health, including muscle composition, power, and energetics…”	Muscle mass, fat mass, power, function, cardiovascular function, metabolic variables	Whole body muscle mass (D_3_-creatine), visceral adipose tissue, abdominal subcutaneous adipose tissue (MRI)	Thigh FFM volume, Thigh muscle fat infiltration (MRI)	Knee extensor peak power (pneumatic), peak power/thigh muscle volume	SPPB, gait speed (400 m)	Physical activity and fitness QoL: MAT-sf questionnaire Mitochondrial energetics: ATPmax, OXPHOS (biopsy) BMI
Engelen et al., 2022 [43]	Normal weight moderate and severe COPD patients, *N* = 32 (18 M:14 F), 66.8 ± 4.4 yrs	“…and improves muscle health (mass and function as secondary outcomes).”	Muscle mass and strength, lung function, and metabolic variables	Whole body and extremity FM and FFM (DXA)		Grip strength		Blood: glucose, C-reactive protein, amino acids, fatty acids, various other health markers Respiratory muscle function: inspiratory pressure, forced expiratory volume, forced vital capacity Physical Activity Scale for the Elderly questionnaire, Saint George Respiratory Questionnaire BMI, waist circumference
Ferguson et al., 2024 [76]	Patients receiving extracorporeal membrane oxygenation, *N* = 23 (10 M:13 F), 48 ± 14 yrs	“…muscle health (size and quality)…”	Muscle size, quality, strength, function, and nutritional data	CSA (US), mCSA (MRI)	Quadriceps thickness and RF echogenicity (US)	Knee extension MVIC (hand-held dynamometer) and muscle strength (Medical Research Council sum score with ICU mobility scale)	Highest level of mobility (ICU mobility scale)	Nutrition data: energy and protein delivery BMI
Finkel et al., 2021 [45]	Males with Duchenne Muscular Dystrophy, *N* = 31, 6.1 ± 1.1 (4–8) yrs	“…lower leg muscle health as determined by the MRI transverse relaxation time constant (T2) from a composite of five muscles.”	T2 relaxation time of lower leg muscles, muscle function, metabolic variables, and gene expression		FF (MRI)		Gait speed (10 m walk/run test), 4 stair climb, time to stand, and North Star Ambulatory Assessment	Blood: cytokine panel of multiple inflammatory markers Gene expression: NF-κB-target genes Heart rate, BMI
Jackson et al., 2022 [49]	Healthy women, *N* = 53, 55.8 ± 5.3 yrs	“…muscle health (muscle mass, grip strength, five-chair rise test, 4 m gait speed test)”: muscle mass, strength, and physical function (i.e., muscle health).”	Muscle mass, strength, function, and dietary intake	SMI (BIA)		Grip strength	Gait speed (4 m walk test) and five-time chair stand test	Intake of energy, protein, carbohydrate, and fat Risk for Sarcopenia BMI
Jacob et al., 2022 [50]	Healthy adults, *N* = 274 (118 M:156 F), 41.9 ± 16.1 (18–70) yrs	“…indices of muscle health should be evaluated in samples of healthy adults to determine the optimum reference values of muscle morphology, function and functional capability.”	Morphology, function, and functional capacity		VL muscle thickness, pennation angle, fascicle length, echo intensity, and contractile properties (US and tensiomyography)	Grip strength	Five-time chair stand test and 1 min chair rise test	Femur length, thigh girth Physical activity level: IPAQ, BPAQ
Locquet et al., 2019 [53]	Adults ≥65 yrs, *N* = 232 (98 M:134 F), 75.5 ± 5.4 yrs (76.0 ± 5.1 yrs M, 75.1 ± 5.6 yrs F)	“Muscle health—SMI (kg/m^2^), grip strength, physical performance…”	Mass, strength, physical performance, nutritional assessment, cognitive assessment, and physical activity	SMI and areal bone mineral density (DXA)		Grip strength	SPPB	Osteoporosis diagnosis: trabecular bone score Mini-Nutritional Assessment Mini-Mental State Examination Self-reported level of physical activity, fracture risk BMI
Olpe et al., 2024 [78]	Patients with cancer, *N* = 269 (161 M:108 F), 68.8 ± 13.3 yrs	“…muscle health markers (i.e., handgrip strength, computed tomography (CT)-based muscle mass and radiodensity)…”	Muscle size, composition, strength, and metabolic variables	Skeletal muscle area, SMI, muscle radiodensity, intramuscular adipose tissue (CT)		Grip strength		Blood: Plasma albumin and c-reactive protein Malnutrition risk BMI
Papaioannou et al., 2021 [55]	Physically active adults, *N* = 191 (69 M:122 F), 67.4 ± 1.5 yrs M, 67.4 ± 1.6 yrs F	“…based on three indicators of muscle health: muscle mass was assessed using bioelectrical impedance and handgrip strength and 5 times sit-to-stand (5-STS).”	Muscle mass, strength, physical function, and dietary intake	SMI (BIA), SMM (Janseen Equation)		Grip strength	Five-time chair stand test	Dietary data: 90-item food-frequency questionnaire, Healthy diet score Adherence to physical activity (Actigraph GT3x) Blood: High-sensitivity c-reactive protein Risk for Sarcopenia
Parker et al., 2021 [56]	Adults during preoperative pancreatic cancer treatment, *N* = 97 (52 M:45 F), 66.4 ± 7.9 yrs	“SMI and SMD were the endpoints of this study; together, they reflect skeletal muscle health.”	Muscle quantity and quality	CSA, SMI—scans performed at T0 and T1 (CT)	SMD (CT)			BMI Risk for Sarcopenia
Pratt et al., 2021 [59]	Healthy older adults, *N* = 300 (150 M:150 F), 64.1 ± 8.5 (50–83) yrs	“…our findings demonstrate the potential of circulating CAF as an accessible indicator of skeletal muscle health in older adults.”	Muscle mass, strength, and metabolic variables	ALM (DXA)		Grip strength		Plasma: CAF Risk of Sarcopenia
Shin et al., 2022 [62]	Adults with chronic kidney disease, *N* = 149 (97 M:52 F), 65 ± 11 yrs	“PhA appears to be a reliable marker for estimating muscle health and HRQoL in patients with CKD.” “…muscle health, inflammatory and muscle-related markers…” “…BIA-derived PhA in estimating the muscle health in patients with CKD. We observed that PhA was related to SMI, handgrip strength, and gait speed; “	Body composition, muscle strength and function, and metabolic variables	FFM, SMM, SMI, intracellular water, extracellular water, and total body water (BIA)		Grip strength	Gait speed (6 m walk test)	Blood: Hemoglobin, albumin, high-sensitivity C-reactive protein, hemoglobin A1c, intact parathyroid hormone, total cholesterol, calcium, phosphorus, sodium, potassium, chloride, total carbon dioxide, blood urea nitrogen, creatinine, and eGFR QoL and risk of Sarcopenia BMI
Song et al., 2022 [63]	Patients who underwent 1-level lumbar microdiscectomy, *N* = 163 (102 M:61 F), 47.8 ± 15.4	“Good” muscle health was defined as score of 2, and “poor” muscle health was defined as score of 0 to 1.” “For the good muscle health group, mean PL-CSA/BMI was 169.4 mm^2^/kg/m^2^, and mean Goutallier class was 1.5.”	Muscle size	Normalized total psoas area (MRI)	Goutallier classification (MRI)			
Song et al., 2023 [83]	Healthy participants with and without a history of spine surgery, *N* = 178 (84 M:94 F), 65.3 ± 12.7 yrs	Muscle health parameters—Goutallier grade, PL-CSA, PL-CSA/BMI, LIV “…novel MRI-based lumbar muscle health grading system incorporating paralumbar cross-sectional areas and Goutallier classification…”	Body size, muscle size, and composition	Paralumbar-CSA, Paralumbar-CSA/BMI ratio, lumbar indentation value (MRI)	Goutallier classification (MRI)			BMI
Su et al., 2022 [64]	Chinese men and women (≥65 years), *N* = 2994 (1424 M:1570 F), 71.9 ± 4.9 yrs	“Our data shows that serum concentrations of individual AAs can be considered biomarkers of muscle health in the older people…”	Body composition, muscle strength and function, and metabolic variables	Lean muscle mass and ALM (DXA)		Grip strength	Gait speed (6 m walk test) and five-time chair stand test	Blood: serum amino acids concentrations Dietary inflammatory index and risk of Sarcopenia BMI
Tan et al., 2022 [65]	Community-dwelling ambulatory older multi-ethnic Asian patients with Type-2 Diabetes Mellitus, *N* = 387 (184 M:164 F), 68.4 ± 5.6 yrs (60–89 yrs)	“…muscle health parameters including muscle mass, strength and gait speed…”	Muscle mass, strength, and function	Muscle mass and SMI (BIA)		Grip strength	Gait speed (6 m walk test)	Physical activity: IPAQ, PASE QoL: World Health Organization Quality of Life scale Systolic and diastolic blood pressures Blood: HbA1c, total cholesterol, HDL, LDL, TG BMI
Vingren et al., 2018 [69]	Men living with Human Immunodeficiency Virus undergoing 60-day inpatient substance abuse treatment, *N* = 16, 42 ± 11 yrs	“…muscle health markers (mass, strength, power).”	Muscle mass, strength, power, and biochemical analysis	Muscle mass estimation (using anthropometric measurements)		Max strength and power (bench press, standing isometric squat)	Vertical jump performance	Blood: IFNγ, IL-1β, IL-2, IL-4, IL-6, IL-10, and tumor necrosis factor (TNF)-α, vascular cell adhesion molecule–1 and cortisol Skinfold thickness, body segment circumferences (upper-arm and forearm)
Virk et al., 2021 [70]	Patients with lumbar spine pathology requiring operation, *N* = 307 (166 M:141 F), 56.1 ± 16.7 yrs	“…muscle health measurements including lumbar indentation value (LIV), paralumbar cross-sectional area divided by body mass index (PL-CSA/BMI), and Goutallier classification of fatty atrophy.”	Muscle size, quality	LIV and PL-CSA/BMI ratio (MRI)	Goutallier classification of fatty atrophy (MRI)			HRQOLs questionnaires: visual analog pain scale back, visual analog pain scale leg, PROMIS scores, Oswestry disability index, short-form 12 mental health score, and short-form 12 physical health score BMI
Virk et al., 2021 [71]	Patients with lumbar spine pathology requiring operation, *N* = 308 (168 M:140 F), 57.7 ± 18.2 yrs	“We measured muscle health by the lumbar indentation value (LIV), Goutallier classification (GC), and ratio of paralumbar muscle cross-sectional area over body mass index (PL-CSA/BMI). A muscle health grade was derived based on whether a measurement showed a statistically significant impact on visual analog scale back and leg pain.”	Muscle size, health related QoL	LIV and PL-CSA/BMI ratio (MRI)	Goutallier classification of fatty atrophy (MRI)			HRQOLs questionnaires: visual analog pain scale back, visual analog pain scale leg, PROMIS scores, Oswestry disability index, short-form 12 mental health score, and short-form 12 physical health score BMI
Yuan et al., 2024 [86]	Older adults in long-term care facilities, *N* = 74 (22 M:52 F), 84.9 ± 7.0 yrs	Muscle health-related indicator: lean mass (SLM, SMM, ASMM, and SMI), handgrip strength, five-time chair stand, and SPPB	Muscle mass, strength, function, and QoL	SLM, SMM, ASM, and SMI (BIA)		Grip strength	Gait speed (6 m walk test), five-time chair stand test, and SPPB	Calf circumference Energy and macronutrient intake QoL
Zhao et al., 2023 [87]	Chinese community-dwelling older women > 65 yrs: *N* = 57, 70.6 ± 4.9 yrs Normal older women: *N* = 10, 70.4 ± 4.4 yrs Older women with pre-Sarcopenia or sarcopenia: *N* = 9, 70.9 ± 3.8 yrs Older women with exercise habits: *N* = 10, 70 ± 3.7 yrs	“In this study, several indicators were selected to reflect muscle health including muscle mass, grip strength, 30 s chair stand, arm curl with a dumbbell, and preferred and maximal gait speed….”	Body size, muscle mass, strength, function	Upper and lower limb skeletal muscle mass and appendicular muscle mass (DXA)		Grip strength	Gait speed (preferred and maximal), chair stand test (30 s), and arm curl reps (2 kg)	BMI
Zhu et al., 2015 [73]	Healthy older postmenopausal women, *N* = 196, 74.3 ± 2.7 yrs	“Over the 2 y, we observed a reduction in the upper arm and calf muscle areas and a decrease in hand-grip strength in women in both the protein and the placebo groups, indicating deterioration in muscle health with aging.”	Muscle mass and function	ASMM (DXA) and upper arm and calf muscle CSA (peripheral quantitative CT)		Ankle dorsiflexion, knee flexor, knee extensor, hip abductor, hip flexor, hip extensor, and hip adductor strength (strain gauge) and grip strength	TUG	Dietary intake, 24 h urinary nitrogen, and levels of physical activity BMI

Abbreviations: AA = Amino acids, ALM = Appendicular lean mass, AMPK = 5′AMP-activated protein kinase, ASMI = Appendicular skeletal muscle index, ASMM = Appendicular skeletal muscle mass, ASR = Absolute synthesis rate, BCAA = Branched-chain amino acid, B-mode = Brightness mode, BIA = Bioelectrical impedance analysis, BMI = Body mass index, BPAQ = Bone physical activity questionnaire, CAF = C-terminal agrin fragment, cESD-10 = Center for epidemiologic studies depression scale, CHAMPS = Community health activities model program for seniors, CKD = Chronic kidney disease, COPD = Chronic obstructive pulmonary disease, COSIAM = Combined oral assessment of muscle, CSA= Cross-sectional area, CT = Computed tomography, DXA = Dual energy x-ray absorptiometry, EAA = Essential amino acids, EQ-5D = 5-level EuroQol, F = Female, F-CSA = Fat cross-sectional area, FF = Fat fraction, FM = Fat mass, FFM = Fat free mass, GL = Lateral gastrocnemius. GM = Medial gastrocnemius, HbA1c = Hemoglobin A1c, HDL = High-density lipoprotein, HRQoLs = Health-related quality of life, IQR = Median with interquartile (25th, 75th percentiles) range, mCSA = Muscle cross-sectional area, ICU = Intensive care unit, IL = Interleukin, IPAQ = International physical activity questionnaire, LDL = Low-density lipoprotein, LIV = Lumbar indentation value, M = Male, MoCA = Montreal cognitive assessment, MPB = Muscle protein breakdown, MPS = Muscle protein synthesis, MRI = Magnetic resonance imaging, MT = Muscle thickness, MVC = Maximum voluntary contraction, MVIC = Maximum voluntary isometric contraction, mHC = Myosin heavy chain protein, MyMHC = Myosin heavy chain gene, NEAA = Sum non-essential amino acids, PASE = Physical activity scale for the elderly, PhA = Phase angle, PL-CSA/BMI = Paralumbar cross-sectional area divided by body mass index, PROMIS = Patient-reported outcomes measurement information system, RF = Rectus femoris, RM = Repetition maximum, SLM = Soft lean mass, SMD = Skeletal muscle density, SMI = Skeletal muscle index, SMM = Skeletal muscle mass, SPPB = Short physical performance battery, QoL = Quality of life, QRT-PCR = Quantitative reverse transcriptase polymerase chain reaction, Sum AA = Sum all measured amino acids, TG = Triglycerides, TNF = Tumor necrosis factor, TUG = Timed up and go test, US = Ultrasound, VO2max = Maximal aerobic capacity, VL = Vastus lateralis, WAnT = Wingate anaerobic test, Yrs = Years.

**Table 2 jfmk-10-00367-t002:** Summary of studies without a definition of ‘muscle health’ included.

Study	Population	Measured	Body Composition	Tissue Composition	Performance	Functional Tasks	Other
Andreo-López et al., 2023 [74]	Adults with type 1 diabetes mellitus, *N* = 62 (21 M:41 F), 38 ± 14 yrs	Body size, composition, strength, and metabolic variables	FFM, FM, total body water, extracellular water, body cellular mass index, SMI, ASMI, and FFM index (BIA)		Grip strength		Blood: Fasting blood glucose, total cholesterol, LDL and HDL cholesterol, triglycerides, albumin, prealbumin, and C reactive protein, glycated hemoglobin A1c, daily total dose insulin, daily total dose insulin per kilogram, and insulin sensitivity factor Lifestyle Parameters: 14-item PREDIMED questionnaire, IPAQ Risk for Sarcopenia BMI
Arentson-Lantz et al., 2019 [32]	Healthy older adults, *N* = 17 (11 M: 6 F), 68 ± 2 yrs	Muscle mass, composition, and metabolic variables	WBLM, WBFM, and LLM (DXA)	CSA and single fiber volume (biopsy with immunohistochemical analysis)	Isokinetic knee extension peak torque (dynamometry)		Dietary intake and step count Blood: blood glucose and plasma insulin (ELISA) BMI
Arentson-Lantz et al., 2019 [33]	Healthy older (60–80 years) adults, *N* = 20 (12 M: 8 F), 68.5 ± 1.5 yrs	Body composition, strength, physical function, and metabolic variables	WBLM, WBFM, and LLM (DXA)		Isokinetic knee extension peak torque (dynamometry)	SPPB and peak aerobic capacity (cycle ergonomic test)	Mean Daily Energy and Macronutrient Intake Blood: blood glucose and serum insulin (ELISA) BMI
Arentson-Lantz et al., 2020 [34]	Healthy older (60–80 years) adults, *N* = 20 (14 M: 6 F), 67.8 ± 1.1 yrs	Body composition, strength, physical function, and dietary intake	WBLM, WBFM, and LLM (DXA)	CSA and single fiber volume (immunohistochemical analysis), protein content—signaling protein expression levels and single fiber characteristics (muscle biopsy—radioimmunoprecipitation assay buffer),	Isokinetic knee extension peak torque (dynamometry)	SPPB and peak aerobic capacity (cycle ergonomic test)	Mean Daily Energy and Macronutrient Intake BMI
Bislev et al., 2019 [37]	Postmenopausal women, *N* = 104, 64.5 yrs (61–68)	Mass, function, physical performance, QoL, and metabolic variables	ALM and FM (DXA)		Maximum voluntary isometric muscle strength, maximum force production (elbow flexion and elbow extension, knee flexion [dynamometry]), and grip strength	TUG, postural stability, and chair rising test	Blood: PTH, 25(OH)D, phosphate, ionized calcium, magnesium, creatinine, and thyroid stimulating hormone Urine: Calcium, phosphate, and magnesium Self-reported physical activity, primary hyperparathyroidism-QoL, and SF36v2 BMI
Bislev et al., 2020 [38]	Healthy postmenopausal women with secondary hyperparathyroidism and vitamin D insufficiency, *N* = 81, 65 (IQR: 61–68.4) yrs	Muscle strength and function, cardiovascular health, and metabolic variables	ASMI and FMI (DXA)		Maximum voluntary isometric muscle strength, maximum force production (elbow flexion and elbow extension, knee flexion [dynamometry]), and grip strength	TUG	Blood: 25(OH)D, 1,25(OH)_2_D, PTH, Ca^2+^, magnesium, phosphate, eGFR, total cholesterol, HDL, LDL, and triglycerides Urine: Creatinine, plasma glucose and lipid profile: hydroxybutyrate, acetate, acetoacetate, acetone, alanine, betaine, carnitine, choline, citrate, creatine, creatinine, dimethylamine, formate, glucose, glutamate, glutamine, glycerol, glycine, isoleucine, lactate, leucine, lysine, methionine, o-phosphocholine, ornithine, phenylalanine, proline, pyruvate, succinate, threonine, trimethylamine n-oxide, tyrosine, urea, valine, τ-methylhistidine Calcium intake Cardiovascular health: blood pressure and arterial stiffness BMI
Cha et al., 2022 [40]	CKD patients, *N* = 150 (97 M: 53 F), 65.0 ± 10.8 yrs	Muscle mass, performance, strength, and metabolic variables	Body composition (BIA)		Grip strength	Gait speed (6 m walk test)	Blood: Indoxyl sulfate, TNF-α, IL-6, myostatin, serum creatinine, eGFR Kidney disease QoL, IPAQ
Engelen et al., 2023 [42]	Moderate to severe COPD patients and healthy controls, *N* = 416 (190 M: 226 F), 68.1 yrs (65.5–71.0)	Muscle mass, strength, respiratory function and metabolic variables	WBFM, extremity FM, FFM, and bone mineral density of spine and hip, ASMI and visceral adipose tissue (DXA)		Maximal leg extension force—one-leg reciprocal extensions (dynamometry), and grip strength.		Blood: Arginine, citrulline, glutamate, glutamine, glycine, histidine, hydroxyproline, isoleucine, leucine, ornithine, phenylalanine, tau-methyl-histidine, taurine, tryptophan, tyrosine, and valine Gynoid to android ratio (DXA) Habitual dietary intake and physical activity level, level of dyspnea, COPD assessment test Respiratory muscle function (hand-held mouth pressure device). BMI
English et al., 2016 [44]	Middle-aged adults, *N* = 19 (12 M: 7 F), 51.5 ± 1 yrs	Muscle mass, function, and quality	WBLM, WBFM, LLM, and body fat percentage (DXA)	Muscle quality (knee extensor peak torque divided by LLM)	Unilateral knee and ankle extensor strength and knee muscle endurance (dynamometer)	Peak aerobic capacity (cycle ergometer)	Dietary intake, Cell signaling and skeletal muscle protein synthesis (muscle biopsy) BMI
Fujie et al., 2024 [90]	Elderly women, *N* = 81, 67.2 ± 5.3 yrs	Muscle mass, quality, strength, and metabolic variables		Quadriceps muscle CSA (MRI), thickness, and echogenicity (US)	1- Repetition Maximum leg extension and biceps curl		Blood: Total cholesterol, HDL, triglycerides, angiotensin II, endothelin-1, complement component 1q, creatinine, and plasma renin activity Blood pressure, heart rate, carotid-femoral pulse wave velocity, carotid β-stiffness
Gil et al., 2022 [46]	Hospitalized COVID-19 survivors *N* = 80 (41 M: 39 F), 59 ± 14 yrs	Muscle strength and size		CSA (US)	Grip strength		Self-perception of health BMI
Granic et al., 2018 [47]	Community-dwelling participants, *N* = 722 (289 M: 433 F), 85+ yrs	Strength, function, protein intake, and physical activity	FM and FFM (BIA)		Grip strength	TUG	Protein intake: 24 h multiple-pass dietary recall Self-reported physical activity BMI
Groenendijk et al., 2020 [48]	Geriatric hip fracture patients, *N* = 40 (11 M: 29 F), 82 ± 8.0 yrs	Muscle mass and strength	ASMM (BIA), muscle thickness (US)		Grip strength		Nutritional status and dietary intake Risk for Sarcopenia
Huang et al., 2023 [77]	Healthy Chinese children 6–9 yrs, *N* = 426 (243 M: 183 F), median 8.0 yrs (IQR = 7.3–8.8 yrs)	Muscle mass, strength, and metabolic variables	ASMM (DXA)		Grip strength		Blood: plasma retinol, plasma ɑ-tocopherol Energy and nutrient intake BMI
Kang et al., 2024 [91]	Elderly adults >60 yrs, *N* = 100 (12 M: 88 F), 65 ± 4 yrs	Muscle strength, physical function, and muscle related hormones	Muscle mass (DXA)		Knee extension torques (isokinetic dynamometry) and grip strength	SPPB, TUG, gait speed	Blood: myostatin, follistatin, and high-sensitivity C-reactive protein
Kang et al., 2024 [92]	Older adults, *N* = 575 (274 M: 301 F), 50–95 yrs	Body composition, muscle and fat mass, strength, and metabolic variables	FM, lean soft tissue, appendicular skeletal muscle mass, visceral adipose tissue, android and gynoid FM ratio (DXA)		Concentric peak torque (isokinetic dynamometer) and grip strength		Blood: amino acid concentrations, C-reactive protein, aspartate, glutamate, hydroxyproline, asparagine, glutamine, citrulline, serine, glycine, arginine, threonine, alanine, taurine, proline, tau-methylhistidine, valine, methionine, isoleucine, leucine, tryptophan, phenylalanine, ornithine, histidine, lysine, tyrosine Respiratory muscle function: Maximal inspiratory pressure PASE and cognitive questionnaire Dietary intake BMI, blood pressure
Kao et al., 2025 [93]	Adults ≥ 65 yrs at risk of malnutrition and sarcopenia, *N* = 97 (24 M: 73 F), 72.4 ± 5.2 yrs	Body composition, strength, function, and metabolic variables	ASM, body fat %, skeletal muscle mass (BIA)		Grip strength	SPPB, 5-time STS, 6 m walk time	Blood: fasting glucose, HbA1c, insulin, homocysteine, creatine, other health measures for cardiometabolic risk factors, renal and liver function SARC-F, SARC-combined with calf circumference, mini nutritional assessment-short form, mini-mental state examination, geriatric depression scale-15 Waist and hip circumference, total body water, BMI
Korzepa et al., 2025 [94]	Healthy middle-to-older adults, *N* = 22 (11 M; 11 F), 61.3 ± 6.5 (50–70) yrs	Body composition, and metabolic variables	Body fat % (DXA)				Blood: plasma glucose, insulin, AA concentration, appetite hormones Respiratory exchange ratio, resting metabolic rate BMI
Lee et al., 2025 [95]	Healthy older adults, *N* = 119 (39 M: 61 F), (65–85) yrs	Body composition, strength, endurance, function, and metabolic variables	Body fat % (BIA)		30 s arm curl test and grip strength	10 m walk test, 30 s STS, TUG, and 3 min incremental step-test	Blood: HbA1c, creatinine, glucose, testosterone, cystatin C, insulin, and measures for liver function, kidney function, blood lipids, and other biomarkers
Li et al., 2021 [51]	Chinese older adults with low lean mass, *N* = 123 (61 M: 62 F), 70 ± 4 yrs	Lean muscle mass, strength and physical performance	ASMI and lean mass (DXA)		Grip strength	SPPB	Daily dietary intake and physical activity level BMI
Locquet et al., 2018 [52]	Community-dwelling older subjects, *N* = 288 (118 M: 170 F), 74.7 ± 5.7 yrs	Muscle mass, strength and physical performance	SMI and areal bone mineral density (DXA)		Grip strength	SPPB	Skeletal status, fracture risk, and risk of Sarcopenia BMI
Matsumoto et al., 2023 [54]	Stroke patients with sarcopenia hospitalized, *N* = 241 (107 M: 134 F), 79.3 ± 10 yrs	Muscle mass, strength, and metabolic variables	SMI (BIA)		Grip strength		Blood: Albumin, c-reactive protein, and hemoglobin Functional independence measure score, ADL assessment, nutritional intake, and risk of Sarcopenia BMI
Peng et al., 2022 [57]	Middle aged and older adults, *N* = 103 (35 M: 68 F), 64.0 ± 8.2 yrs	Muscle size, composition, strength, performance, and metabolic variables	Total FM and FFM (BIA), and relative ASMM (MRI)	IMAT and CSA (MRI)	Grip strength	Gait speed (6 m walk test)	Blood: Serum albumin, alanine aminotransferase, uric acid, total cholesterol, HDL, LDL, triglyceride, serum creatinine, high-sensitivity C-reactive protein, and fasting glucose; Whole blood glycated hemoglobin Cognitive function, nutritional and mood status IPAQ, BMI
Peng et al., 2024 [79]	Adults with inadequate protein intake, *N* = 97 (18 M: 79 F), 64.7 ± 4.8 yrs	Muscle size, strength, physical function, metabolic variables and quality of life	Relative ASMM (BIA)	Body fat percentage (BIA)	Grip strength	Usual gait speed (6 m), 6 min walk test, and five-time chair stand test	Blood: Albumin, creatinine, alanine aminotransferase, total cholesterol, HDL, LDL, uric acid, fasting glucose, dehydroepiandrosterone sulfate, insulin-like growth factor-1, homocysteine, high-sensitive c-reactive protein, vitamin D3, glycated hemoglobin, myostatin, and leptin Cognition: MoCA, CES-D, IPAQ Nutritional status SF-36, BMI
Pérez-Piñero et al., 2021 [58]	Caucasian men and postmenopausal women, *N* = 45 (8 M: 37 F), 58.9 ± 6.1 (50–75) yrs	Muscle mass, function, strength, quality, and metabolic variables	FM, lean mass, muscle mass, and ASMM (DXA)	Muscle quality (muscle mass between the peak torques)	Knee extension torques (isokinetic and isometric dynamometry) and grip strength		Blood pressure, health-related QoL, SF-36, dietary intake BMI
Raghupathy et al., 2023 [80]	Adults and children, *N* = 962 (428 M: 534 F), 60 ± 9 (5–70) yrs	Body size, muscle composition, quality, strength, physical activity level, and blood markers of inflammation	ALM (DXA), subcutaneous and visceral adipose tissue (CT)	Upper extremity muscle quality (strength per kilogram of lean mass)	Knee extension (hand-held isometric dynamometry) and grip strength		Blood: IL-6, monocyte chemoattractant protein-1, resistin, and adiponectin (ELISA) Physical activity BMI
Rousseau et al., 2015 [60]	Adults with thermal burns, *N* = 15 (11 M: 4 F), 50 (25–64) yrs	Muscle strength and metabolic variables	Bone mineral density (DXA)		Knee muscle strength (isokinetic dynamometry)		Blood: 25OH–D, 1,25(OH)2–D, calcium, fibroblast growth factor 2, PTH, phosphate, creatine, collagen type 1 cross-linked C-telopeptide, serum type 1 procollagen N-terminal and serum bone alkaline phosphatase
Sabir et al., 2023 [81]	Norwegian adults, *N* = 1317 (578 M: 739 F), 67–70 yrs	Muscle mass, body composition, strength, physical activity, and habitual dietary intake	SMM, ASMM, ASMI, total body FM and percentage (BIA)		Grip strength		Habitual dietary intake Self-reported physical activity BMI
Schneider et al., 2015 [61]	Healthy adults in microgravity environments, *N* = 11 (9 M: 2 F), 40 ± 7 yrs	Mechanical properties of skeletal muscles and tendons					Oscillation frequency (Hz), dynamic stiffness (N/m), elasticity, mechanical stress relaxation (ms) time, creep (Deborah number) (MyotonPRO device) BMI
Seo et al., 2024 [82]	Healthy adult golfers, *N* = 57 (27 M: 30 F), ~59 ± 9.5 (26–64) yrs	Body size, body composition, muscle strength, golf performance, physical function, and metabolic variables	SMM and FM (BIA)		Knee extension and flexion strength (dynamometry) and grip strength	Golf drive distance, club-head speed, ball speed, 2 min push-up test, and MFT balance test	Blood: lactic acid, creatine, lactate dehydrogenase, creatine kinase, blood urea nitrogen, red blood cell, white blood cell, hemoglobin, platelet, hematocrit, glucose, aspartate aminotransferase, alanine transaminase, and gamma-glutamyl transferase Dietary intake and levels of physical activity Blood pressure, heart rate, BMI
Van Ancum et al., 2020 [66]	Community-dwelling adults, *N* = 197 (57 M: 140 F), 67.9 (57–75.1) yrs	Body composition, muscle mass, strength, and function	SMM, SMI, ALM, ALM/height^2^, SMM and ALM relative to body weight (BIA)		Grip strength	Gait speed (4 m walk test)	Self-reported levels of physical activity, ADL, and risk of Sarcopenia BMI
Van Dongen et al., 2020 [67]	Community-dwelling older adults, *N* = 168, (66 M:102 F), 75 ± 6 yrs	Body composition and mass, muscle strength and function	Lean body mass, ALM, and FM (DXA)		Lower limb 3-Repetition Maximum test (leg press and leg extension machines) and knee extension strength (dynamometry)	Gait speed (6 min walk test and 4 m walk test), SPPB, and TUG	QoL, ADL, nutritional status, dietary intake, and risk of Sarcopenia BMI
Vesey et al., 2020 [68]	Children and adolescents with conditions that impacted musculoskeletal health, *N* = 17, 15.7 ± 2.9 yrs	Body composition and function	Whole body: FM, lean mass, bone mineral content, and bone mineral density Lumbar spine: bonce mineral content and bone mineral density (DXA)			Gait speed (6 min walk test), chair stand test, balance test, and single leg jump test	BMI
Vitale et al., 2020 [72]	Healthy older adults, *N* = 9 (3 M: 6 F), 68 ± 7 (62.9–73.1) yrs	Body composition, muscle strength and function	Lean mass, FM, ASMI (DXA) and CSA of thigh (MRI)		Maximum isometric strength of knee flexor and extensor (dynamometry) and grip strength	Chair stand test (30 s) and Mini balance evaluation systems test	BMI
Xiong et al., 2024 [84]	Older adults with high fall risk, *N* = 160, 68.5 ± 8.9 (65–85) yrs	Muscle mass and function	Bone mineral density and lower limb muscle mass (DXA)			Berg balance scale, TUG, chair stand test (30 s), and fall-risk assessment tool	Fall-risk questionnaire
Yoshimura et al., 2024 [85]	Stroke patients, *N* = 955 (511 M: 443 F), 73.2 ± 13.3 yrs	Muscle mass, strength, and metabolic variables	SMI (BIA)		Grip strength		Blood: Albumin, hemoglobin, c-reactive protein Energy and protein intake and pre-stroke ADL

Abbreviations: 25(OH)D = 25-hydroxy vitamin D, 1,25(OH)2D= 1,25dihydroxy vitamin D, AA = Amino acids, ADL = Activities of daily living, ALM = Appendicular lean mass, ASMI = Appendicular skeletal muscle index, ASMM = Appendicular skeletal muscle mass, BIA = Bioelectrical impedance analysis, BMI = Body mass index, Ca^2+^ =Ionized calcium, CESD-10 = Center for epidemiologic studies depression scale, CKD = Chronic kidney disease, COPD = Chronic obstructive pulmonary disease, CSA = Cross-sectional area, CT = Computed tomography, DXA = Dual energy x-ray absorptiometry, eGFR = Estimated glomerular filtration rate, ELISA = Enzyme-linked immunosorbent assay, F = Female, FM = Fat mass, FMI = Fat mass index, FFM = Fat free mass, HDL = High-density lipoprotein, IL = Interleukin, IMAT = Intramuscular adipose tissue, IPAQ = International physical activity questionnaire, LDL = Low-density lipoprotein, LLM = Leg lean tissue mass, M = Male, MoCA = Montreal cognitive assessment, MRI = Magnetic resonance imaging, PTH = Parathyroid hormone, QoL = Quality of life, SF-36 = Short form-36 health survey, SMI = Skeletal muscle index, SMM = Skeletal muscle mass, SPPB = Short physical performance battery, TNF = Tumor necrosis factor, TUG = Timed up and go test, US = Ultrasound, WBFM = Whole body fat mass, WBLM = Whole body lean tissue mass, Yrs = Years.

A word cloud of the 29 operational definitions [31,35,36,39,41,43,45,49,50,53,55,56,59,62,63,64,65,69,70,71,73,75,76,78,83,86,87,88,89], is provided in Figure 4. Operational definitions most commonly included ‘muscle mass’ (11), ‘grip-strength’ (9), ‘cross-sectional area’ (7), ‘function’ (6), ‘strength’ (6), ‘power’ (4), ‘gait speed’ (4), ‘skeletal muscle index’ (4), ‘Goutallier’ classification (4), ‘size’ (3), ‘quality’ (2), ‘physical performance’ (2), ‘mass’ (2), ‘phase angle’ (2), ‘lumbar indentation’ (2), and ‘chair stand’ (2).

**Figure 4 jfmk-10-00367-f004:**
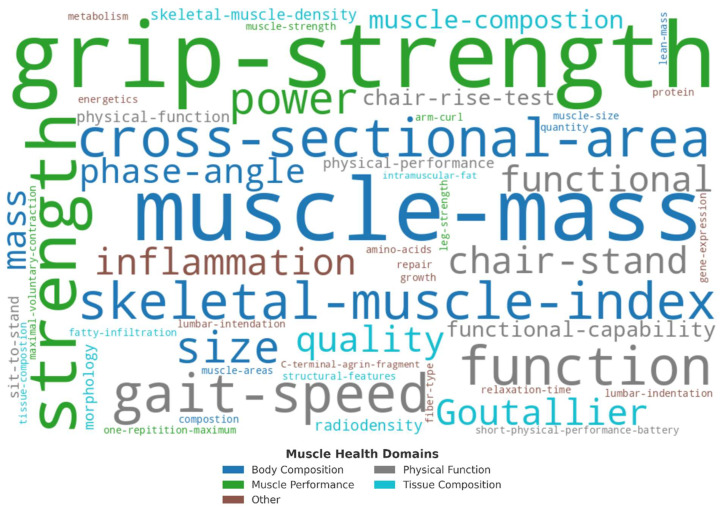
Word cloud visualization of key words extracted from 31 operational definitions of muscle health. Words were categorized into five components: body composition (blue), physical function (gray), muscle performance (green), tissue composition (teal), and other (rust). Word size reflects term frequency across definitions.

Of the 29 studies providing an operational definition [31,35,36,39,41,43,45,49,50,53,55,56,59,62,63,64,65,69,70,71,73,75,76,78,83,86,87,88,89], all but one [45] (*n* = 28, 96.6%) assessed body composition or muscle size, 20 (69%) measured muscle performance (e.g., grip strength, isometric or isokinetic strength) [35,39,43,49,50,53,55,59,62,64,65,69,73,75,76,78,86,87,88,89], 18 (62.1%) measured functional performance (e.g., short physical performance battery [SPPB], gait speed) [35,41,45,49,50,53,55,62,64,65,69,73,75,76,86,87,88,89], while 13 (44.8%) included tissue composition (e.g., echogenicity, intramuscular adipose tissue) assessments [31,35,36,39,50,56,63,70,71,78,83,88,89]. The 36 studies that used, but did not define ‘muscle health’, had a similar assessment distribution to the studies that provided an operational definition. Most studies have emphasized body composition or muscle mass (*n* = 31, 86.1%) [32,33,34,37,38,40,42,44,47,48,51,52,54,57,58,66,67,68,72,74,77,79,80,81,82,84,85,92,93,94,95], and muscle performance (*n* = 31, 86.1%) [32,33,34,37,38,40,42,44,46,47,48,51,52,54,57,58,66,67,72,74,77,79,80,81,82,85,90,91,92,93,95], with fewer incorporating functional tasks (*n* = 23, 63.9%) [33,34,37,38,40,42,44,47,51,52,57,60,66,67,68,72,79,82,84,85,91,93,95], or tissue composition (*n* = 7, 19.4%) [32,34,44,46,57,58,90].

The frequency of defined and inferred ‘muscle health’ measures across all 65 identified studies is summarized in Figure 5. Sixty studies (92.3%) measured body composition in some way (e.g., total body fat percentage, appendicular lean mass) [31,32,33,34,35,36,37,38,39,40,41,42,43,44,45,47,48,49,50,51,52,53,54,55,56,57,58,59,62,63,64,65,66,67,68,69,70,71,72,73,74,75,76,77,78,79,80,81,82,83,84,85,86,87,88,89,92,93,94,95], 51 (78.5%) assessed muscle performance (e.g., grip strength, isometric force) [32,33,34,35,37,38,39,40,42,43,44,46,47,48,49,50,51,52,53,54,55,57,58,59,62,64,65,66,67,69,72,73,74,75,76,77,78,79,80,81,82,85,86,87,88,89,90,91,92,93,95], and 41 (63.1%) examined physical function (e.g., timed up-and-go [TUG], balance) [33,34,35,37,38,40,41,42,44,45,47,49,50,51,52,53,55,57,60,62,64,65,66,67,68,69,72,73,75,76,79,82,84,85,86,87,88,89,91,93,95], while 20 (30.8%) included at least one measure of tissue composition (e.g., echogenicity, intramuscular adipose tissue) [31,32,34,35,36,39,44,46,50,56,57,58,63,70,71,78,83,88,89,90]. Other common assessments included BMI (47, 72.3%) [31,32,33,34,37,38,41,42,43,44,45,46,47,49,51,52,53,54,56,57,58,61,62,64,65,66,67,68,70,71,72,73,74,75,76,77,78,79,80,81,82,83,87,88,89,93,94], metabolic biomarkers (*n* = 31, 47.7%) [32,33,37,38,39,40,42,43,45,54,55,57,58,59,60,62,64,65,69,74,77,78,79,80,82,88,90,91,93,94,95], dietary/nutritional tracking (*n* = 23, 35.4%) [32,33,34,35,38,44,47,48,49,51,53,54,55,64,67,73,76,77,78,81,82,85,86], activity, quality of life, and pain questionnaires (*n* = 27, 41.5%) [36,37,40,41,42,43,46,47,50,51,53,57,58,62,65,66,67,70,71,74,75,79,80,81,82,84,89].

Nearly all studies (*n* = 61, 93.8%) included more than one ‘muscle health’ component [31,32,33,34,35,36,37,38,39,40,41,42,43,44,46,47,48,49,50,51,52,53,54,55,56,57,58,59,61,62,64,65,66,67,68,69,70,71,72,73,74,75,76,77,78,79,80,81,82,83,84,85,86,87,88,89,90,91,92,93,95]. The distribution of this is illustrated in Figure 6. Only five studies (7.7%) included all four primary ‘muscle health’ components [34,35,50,57,88]. The most common combination (*n* = 27, 41.5%) included body composition (e.g., muscle mass, body fat %), muscle performance (e.g., grip strength, knee extension torque), and physical function (e.g., TUG, sit-to-stand) [33,38,40,42,43,49,51,52,53,55,62,64,65,66,67,69,72,73,75,76,79,82,85,86,87,93,95]; followed by body composition and muscle performance (N = 8, 12.3%) [48,54,59,74,77,80,81,92], and body and tissue composition (*n* = 6, 9.2%) [31,36,56,63,83,89].

The methods of assessing body and tissue composition varied (Figure 7), with dual-energy X-ray absorptiometry (DXA) being the most used tool (*n* = 29, 44.6%) [32,33,34,35,37,38,39,41,42,43,44,51,52,53,58,59,60,64,67,68,72,73,77,80,84,87,91,92,94], followed by bio-electrical impedance (BIA) (*n* = 19, 29.2%) [40,47,48,49,54,55,57,62,65,66,74,79,81,82,85,86,88,93,95], magnetic resonance imaging (MRI) (*n* = 12, 18.5%) [31,36,45,57,63,70,71,72,75,76,83,89], ultrasound (US) (*n* = 8, 12.3%) [35,39,46,48,50,76,88,90], tissue biopsy (*n* = 5, 7.7%) [31,32,34,35,39], and CT (*n* = 4, 6.2%) [56,73,78,80].

Muscle performance was measured using various methods (Figure 8). The most frequently used test was grip strength (*n* = 42, 64.6%) [35,37,38,39,40,42,43,46,47,48,49,50,51,52,53,54,55,57,58,59,62,64,65,66,72,73,74,75,77,78,79,80,81,82,85,86,87,88,91,92,93,95], followed by knee extension (*n* = 21, 32.3%) [32,33,34,35,39,42,43,44,58,60,67,72,73,75,76,80,82,89,90,91,92], and flexion (*n* = 4, 6.2%) [37,38,72,73] force, torque, or power. A few studies utilized elbow and flexion strength (*n* = 2, 3.1%) [37,38], bench press and squat strength (*n* = 1, 1.5%) [69], or the strength of other single muscle groups (*n* = 5, 7.7%), including ankle dorsiflexion, hip abductor, hip flexor, hip extensor, and hip adductor strength [36,44,67,73,76]. Other studies assessed respiratory muscle functions (e.g., inflationary pressure; *n* = 3, 4.6%) [42,43,92], and electrical stimulation relaxation times (*n* = 1, 1.5%) [45].

Methods used to assess physical function also varied widely (Figure 9), with gait speed (e.g., typical pace, maximal speed, time to a set distance) being the most common (*n* = 18, 27.7%) [40,45,49,57,62,64,65,66,67,68,75,79,86,87,89,91,93,95]. Other common tests included the SPPB (*n* = 14, 21.5%) [33,34,39,51,52,53,67,75,84,86,88,89,91,93], sit-to-stand/chair rise variations (*n* = 13, 20%) [37,49,50,55,64,68,72,79,84,86,87,93,95], and TUG variations (*n* = 9, 13.8%) [37,38,41,47,67,73,84,91,95]. A few studies also employed balance tests (*n* = 6, 9.2%) [37,68,72,75,82,84], while power was assessed via Wingate (*n* = 4, 6.2%) [33,34,35,44], vertical jump (*n* = 2, 3.1%) [68,69], and sprint (*n* = 1, 1.5%) tests [35]. Ten (15.3%) studies employed other single measures of physical function, such as self-reported physical activity levels or fatigue [42,45,75,76,82,84,85,87,89,95].

## 4. Discussion

While the term ‘muscle health’ is widely used, definitions, applications, and measurement methods vary greatly across the literature. Using a proposed framework for muscle health informed by the ICF, we conducted a narrative review to better understand the operational definitions of the term in the literature and synthesize these usage patterns where possible and appropriate. Overall, 65 studies were identified, with 29 providing an operational definition of ‘muscle health’. An additional 36 studies used the term ‘muscle health’ but did not provide an operational or conceptual definition. From the 65 studies, we characterized the study sample and outcome measures associated with muscle health categorized by their measurement domains: body/muscle tissue composition, muscle performance, and functional status. A key limitation across the studies that used but did not define ‘muscle health’ is the lack of conceptual clarity. ‘Muscle health’ was often used interchangeably with sarcopenia, muscle quality, or general musculoskeletal status, without justification for why particular measures were included or excluded. This inconsistency undermines comparability across studies, as similar outcomes were variably treated as either central or peripheral to “muscle health.” Furthermore, reliance on surrogate markers such as BMI or broad functional questionnaires, without integration into a clear conceptual framework, makes it difficult to interpret whether observed associations truly reflect skeletal muscle status. These limitations underscore the importance of establishing a standardized framework, so that “muscle health” assessments can be applied consistently in both clinical and research contexts.

### 4.1. Common Elements of Muscle Health

Body composition (e.g., muscle mass, fat percentage, appendicular lean mass) was measured in 92.3% of the studies. Nevertheless, the definitions of muscle health were variable across the selected studies. Twenty-nine of the 65 studies defined muscle health by listing associated outcome measures such as muscle mass, grip strength, and physical function (e.g., gait speed, chair stand test, TUG). The lack of consensus was reflected in many studies that featured indirect outcome measures, such as BMI (73.9%) and metabolic biomarkers (47.7%), as components of muscle health. Notably, 93.8% of the reviewed studies integrated multiple outcomes, with 60% of the publications including at least three components of muscle health. The measurement domains in our proposed muscle health framework (i.e., body/muscle tissue composition, muscle performance, and functional status) were present in 49.2% of the reviewed studies.

Body composition, particularly muscle mass, has long been considered a cornerstone of muscle health. Our findings showed that the methods used to assess body and tissue composition varied throughout the literature. Ultrasound is emerging as a method for estimating body/muscle tissue composition, which is used more frequently than tissue biopsy and CT imaging. Nonetheless, DXA, BIA, or MRI were used in 92.3% of the studies. The variability in these measurement methods reflects the competing needs of accommodating available clinical and research resources with the effort to establish standardized approaches across studies. Given the importance of assessing muscle in patient settings that may range from community-based clinics to large medical centers, a stratified approach to evaluate muscle health must be considered. An analogous approach to musculoskeletal disorders has been adopted by organizations such as the American College of Rheumatology and the European Alliance of Associations for Rheumatology, which provide guidelines for diagnosing rheumatic conditions, both with and without laboratory values [96]. In a similar vein, characterizing the body/muscle tissue composition domain of muscle health may incorporate bioimaging devices ranging from ultrasound to MRI, depending on equipment access, cost limitations, and the complexity of the clinical environment.

The primary use of methods designed to estimate lean body mass (DXA: 44.6% and BIA: 29.2%), rather than specifically muscle mass, poses challenges to assessing muscle health. Bioimaging methods such as DXA, that estimate lean body mass as a surrogate measure of muscle mass, include a significant proportion of non-contractile tissue (i.e., approximately 25% of skin and connective tissue) [97]. In addition, DXA estimates of lean body mass often have low associations with frailty outcomes [98,99,100] and are less responsive to post-exercise regimen changes compared to local measures of muscle size, as measured via CT or MRI [101,102,103]. The extensive use of DXA in previous studies and its availability in hospital settings have been cited as reasons to maintain this bioimaging modality as a “reference” standard device and to continue using lean body mass as a component of muscle health [104]. Nevertheless, no current non-invasive method offers an exact quantification of skeletal muscle mass, with each approach, including DXA, BIA, skinfolds, CT, MRI, ultrasound, and even emerging techniques such as D3-creatine, carrying inherent assumptions and limitations [104]. Accordingly, these methods should be viewed as providing useful, but imperfect proxies, each with unique strengths and drawbacks. However, contemporary reappraisals of this approach have noted that techniques such as D3-creatine may provide a more accurate estimate of whole-body muscle mass, and that bioimaging methods, including MRI, CT, and ultrasound, offer estimates of both muscle mass and tissue composition [105,106,107,108,109]. Consequently, the high frequency of DXA and other methods of lean body mass assessment cited in the reviewed studies may be an insufficient rationale to continue this methodological approach in future studies of muscle health. Additionally, the role of tissue composition (such as the extent of fatty infiltration in muscle) emerged as a significant factor influencing muscle health, suggesting that future definitions and assessments should integrate both mass and tissue quality [2].

Muscle performance is an essential domain of muscle health, as evident from the various strength assessment methods employed in these studies. Grip strength was the most frequently used technique (64.6%) to assess muscle performance, demonstrating its ease of use, portability, and presumed utility as a surrogate measure of whole-body strength. While the use of grip strength is limited by its low-to-moderate association with lower extremity strength [110,111], it remains an important outcome measure in field studies involving older adults due to its low testing burden and well-known psychometric properties [20,112]. Knee extension strength was the second most measured aspect of muscle performance (32.3%). Lower extremity muscle performance has a stronger relationship with physical functioning, such as gait speed, in comparison to upper extremity strength [110]. Overall, the strong association between muscle performance and mobility, as well as hospitalization risk, emphasizes its relevance as a predictor of health [113]. The findings of the current study support the inclusion of muscle performance as a standard part of muscle health assessments. Specific testing methods and muscle groups used to characterize muscle performance may vary depending on the availability of equipment, the population of interest, and the rationale for assessment (e.g., general screening versus an assessment of specific muscle groups).

Functional status is a crucial aspect of health-related quality of life, with gait speed being the most used method (27.7%) to characterize this domain of muscle health in the reviewed studies. Gait speed is a strong predictor of health outcomes such as mortality and hospitalization, and is a low-burden assessment, making it ideal for both research and clinical settings [114]. However, there are many variations in the methods used for testing gait speed (e.g., speed, distance, customary or fastest gait speed). A previous study involving older adults with muscle dysfunction revealed that individuals with significant lower extremity strength deficits may still maintain walking speeds that exceed 1.0 m/s [110]. More demanding functional tasks, such as one’s fastest walking speed [114], may show a stronger association with muscle strength in comparison to customary walking speed [110]. While variation in the testing method for gait speed allows assessment flexibility, this approach can also lead to methodological inconsistencies across studies. Following gait speed, the SPPB (21.5%), chair rise tests (20%), and TUG (13.8%) were the widely used functional assessments in the reviewed studies. These methods provide meaningful information on lower limb strength, balance, and overall mobility, which can directly impact ADL. By combining selected functional tasks through assessment batteries, such as the SPPB, one can obtain a comprehensive assessment of functional status, reflecting an individual’s ability to perform these mobility-related activities. Nevertheless, the multi-system contributions to functional status require an appropriate patient history and physical exam to determine if muscle dysfunction is a key contributor to observed functional limitations and diminished mobility. Additionally, functional tests vary in their relative difficulty and bias towards either muscle strength or power. For example, tasks with a focus on muscle power, such as the 30 s chair rise test, may reveal performance deficits earlier than less demanding tasks, such as usual gait speed [115]. An additional point of consideration is that diminished muscle health is often found in people with chronic conditions who are non-ambulatory or have other functional limitations [6]. Consequently, alternative methods to assess the functional domain of muscle health in adults with disabilities merit additional study.

### 4.2. Implications for Muscle Health Assessment

The assessment of muscle health has important implications for various patient populations, including older adults with sarcopenia and those with chronic health conditions [116,117,118]. Determining a viable model for muscle health and consistent measurement domains can ensure a more comprehensive evaluation of muscle health, aiding in the detection of early muscle loss or diminished quality in those at risk for muscle dysfunction. A proactive approach to screening or evaluating muscle-related impairments can help mitigate adverse outcomes, such as decreased independent mobility and compromised health-related quality of life. However, the findings from the current work revealed variability in the definitions and measurements of muscle health across studies, highlighting the need for consensus development and the establishment of standardized assessment guidelines. While 31 of the reviewed studies provided operational definitions of muscle health, it is essential to note that these definitions primarily served as documentation of muscle-related outcome measures. Rarely are frameworks or conceptual definitions provided or cited to provide a rationale for the collection of muscle-related outcomes featured in the reviewed study methods.

There have been notable recent efforts to standardize approaches to muscle-related outcome measures and provide a rationale for identifying components that characterize muscle health [5,119,120]. The Global Leadership Initiative in Sarcopenia (GLIS) has addressed competing definitions of sarcopenia and conducted an international Delphi Study to move toward a common classification approach [120,121]. The findings from the Delphi process indicated that three components of sarcopenia should comprise the conceptual definition of the condition: muscle mass (89.4%), muscle strength (93.1%), and muscle-specific strength (80.8%) [120]. While it could be argued that the efforts of the GLIS investigators are limited explicitly to sarcopenia, their recommendation to include measures of both muscle mass and strength is consistent with the proposed muscle health measurement domains for body/muscle tissue composition (muscle mass) and muscle performance (muscle strength and muscle-specific strength). Moreover, their identification of muscle-specific strength (e.g., strength standardized to muscle size or other scaling factors) raises an important point about strength assessment methodology. The studies featured in the review included standardized measures of strength assessment. Nonetheless, additional empirical findings and consensus efforts may inform the relative value of expressing muscle performance in terms of peak torque, work, power, and relative peak torque scaled to body stature or muscle size.

Heymsfield and colleagues [5] have also addressed the challenge of characterizing muscle health. The investigators note that form (e.g., body/muscle tissue composition) and functional measures are often framed as equivalent criteria in clinical decision-making algorithms. Instead, the classic biological concept of “function follows form” provides a hierarchy informed by the pathophysiological links between muscle characteristics and clinical outcomes [5]. A classification system informing the proposed muscle health framework in the current study is the ICF, which encompasses domains of ‘Body Functions’, ‘Structures’, ‘Activities’, and ‘Participation’ [18]. While the ICF is not based on a hierarchical model as proposed by Heymsfield and associates [5], there is consistency between the proposed domains of muscle health identified in this study (body/muscle tissue composition, muscle performance, and functional status) and elements of Heymsfield et al.’s “Outcomes Follow Function Rule” (form, function, and outcomes) [5,18]. The key difference between these conceptual approaches is that the recommendation in the current work categorizes direct measures of muscle performance separately from functional performance tasks such as gait speed or chair stands, given that body systems beyond the musculoskeletal system impact functional status. In contrast, Heymsfield et al. [5] categorize both muscle performance and functional status within the domain of “function” and distinguish between “outcomes” as global assessments of morbidity and mortality. Overall, the domains of muscle health proposed in this work are well-supported by existing frameworks for assessing physical health [27], consensus-based component measures [5,119,120], and the most frequently cited measures in the reviewed studies (Figure 10).

### 4.3. Toward a Standardized Approach to Assessing Muscle Health

While this review highlights substantial variability in definitions and measurement methods, consistent domains emerged across studies that align with existing consensus recommendations in sarcopenia and physical function research. Based on frequency of use, psychometric strength, and feasibility, we propose that the identified domains of (1) Body Systems/Structures (i.e., body/muscle tissue composition and muscle performance) and (2) Participation (i.e., functional status) provide a foundation to develop assessment guidelines for clinical and research applications (see Figure 10). Assessment tools corresponding with these domains have well-documented associations with health outcomes, hold solid psychometric properties, and can be implemented across a range of settings, reflecting the multidimensionality of muscle health.

The use of simple standardized assessment tools in clinical settings does not preclude the adoption of more advanced measures in research settings (e.g., grip strength testing versus isokinetic dynamometry). While advanced assessment tools for muscle performance or tissue characteristics are appropriate in specialized contexts, accounting for the availability of both simple and advanced methods will facilitate the creation of a practical roadmap towards standardized assessment guidelines. In clinical settings, prioritizing feasible and validated assessment tools (e.g., grip strength, gait speed) ensures broad applicability. In research contexts, the scope may be expanded to include more detailed compositional and performance-based measures to aid mechanistic research. Importantly, a practical roadmap towards standardized assessment guidelines for muscle health includes addressing the issues of data acquisition and interpretation. This effort encompasses a range of tasks, from addressing data processing issues and developing specific protocols for performance-based tests to normalizing strength measurements based on body size or muscle volume. Gaining clarity on data acquisition and interpretation issues related to assessing muscle health will require further consensus efforts and additional methodological studies. Nonetheless, this tiered approach to standardized assessment methods across practice settings may improve comparability across studies while maintaining flexibility for both practitioners and investigators.

### 4.4. Limitations

Despite the comprehensive nature of this review, several limitations must be acknowledged. First, many studies inferred definitions of muscle health through outcomes without explicitly defining the term. Second, by restricting our search to studies that explicitly used the term “muscle health”, we may have excluded research employing closely related constructs (e.g., “muscle quality,” “sarcopenia,” or “muscle function”); however, this was a deliberate methodological decision to examine how the specific term “muscle health” is currently defined and operationalized. Given the search criteria employed in this work, comparing “muscle health” to related concepts such as “muscle quality” was beyond the scope of this narrative review. Furthermore, reliance on specific databases (only CINAHL and PubMed) may have introduced bias in the selection of studies, potentially overlooking pertinent research published elsewhere. In addition, heterogeneity in study design and participant samples makes generalizing the findings across all demographic groups challenging. Most importantly, although we conducted a systematic search to assess the current literature, the overarching narrative format of this review is susceptible to bias due to the authors’ perspective in the manuscript. Thus, our viewpoints are not infallible, and this paper is open to further and differing interpretations. Lastly, our review focused primarily on skeletal muscle health, which limits the generalizability of our findings to other muscle types, such as cardiac or smooth muscle.

## 5. Conclusions

This narrative review underscores the complexity of defining and assessing muscle health. While muscle mass remains a crucial outcome measure, muscle health is a multifaceted concept that encompasses not only muscle mass but also muscle performance, tissue composition, and physical function. As such, readers can, and likely should, interpret ‘muscle health’ as a term that is informed by general and physical health. Furthermore, these concepts can include muscle morphology and morphometry, muscle performance, and functional impairments and limitations, as observed in 49.2% of the selected studies for review. The muscle health domains recommended in this work are consistent with established frameworks for assessing physical health [27] and the ICF model to classify components of health and well-being [18]. The need for standardized definitions and consensus-based guidelines is evident, as is the importance of considering these elements in varied clinical and research settings. Healthcare providers can better manage the risks associated with muscle dysfunction and improve patient outcomes by adopting a holistic and proactive approach to assessing muscle health.

## Figures and Tables

**Figure 1 jfmk-10-00367-f001:**
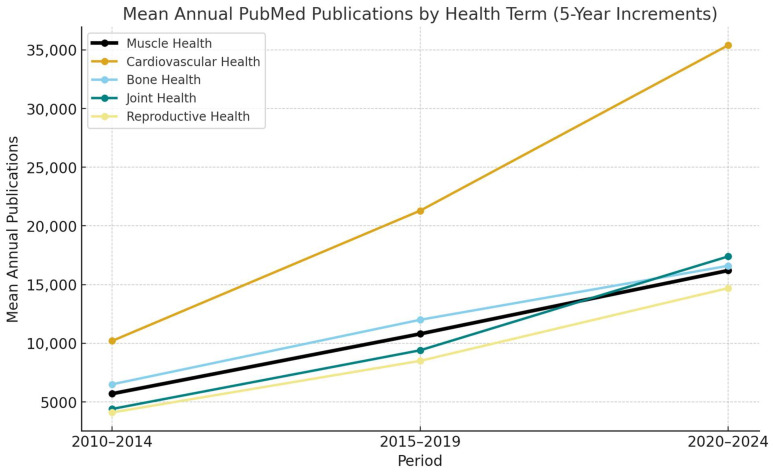
Absolute rise in PubMed database hits across multiple health types from 2010 to 2024 in 5-year increments.

**Figure 2 jfmk-10-00367-f002:**
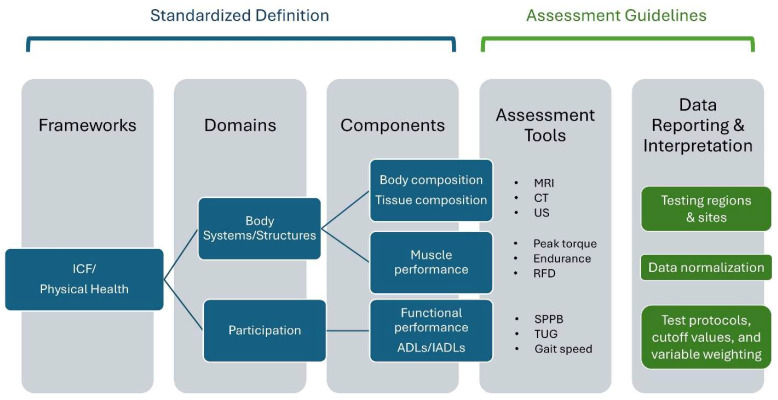
The framework for ‘muscle health’ reflects the multidimensional aspects of general health, physical health, and physical performance. This approach is grounded in the physical dimension of health using the classification system of the International Classification of Functioning, Disability and Health (ICF) developed by the World Health Organization. This framework for muscle health includes the ICF health-related domains: (1) Body Systems/Structures and (2) Participation. The components of these domains represent categories of assessment: (1) body/muscle tissue composition, (2) muscle performance, and (3) functional performance. Each muscle health component may be quantified using various assessment tools (selected tests are listed for illustrative purposes). Guidelines concerning testing protocols and data interpretation impact the use of assessment tools to characterize muscle health. ADL: activities of daily living; IADL: instrumental activities of daily living; MRI: magnetic resonance imaging; CT: computed tomography; US: ultrasound; RFD: rate of force development; SPPB: short physical performance battery; TUG: timed up-and-go.

**Figure 3 jfmk-10-00367-f003:**
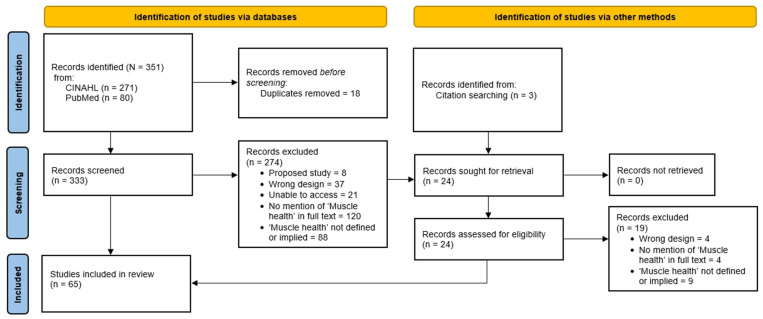
Preferred Reporting Items for Systematic Reviews and Meta-Analyses (PRISMA) flow chart.

**Figure 5 jfmk-10-00367-f005:**
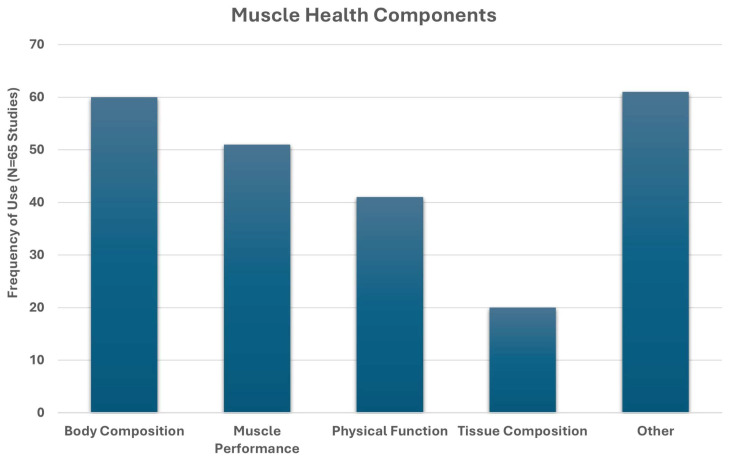
Outline of identified ‘muscle health’ definitions included in articles (*N* = 65) obtained via search and screenings.

**Figure 6 jfmk-10-00367-f006:**
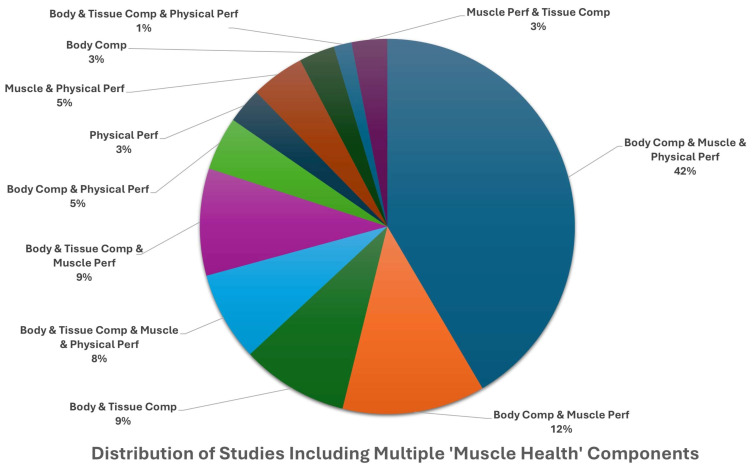
Frequency of combined ‘muscle health’ components featured as outcome measures across all studies (*N* = 65) included in the review. Comp: composition; Perf: performance.

**Figure 7 jfmk-10-00367-f007:**
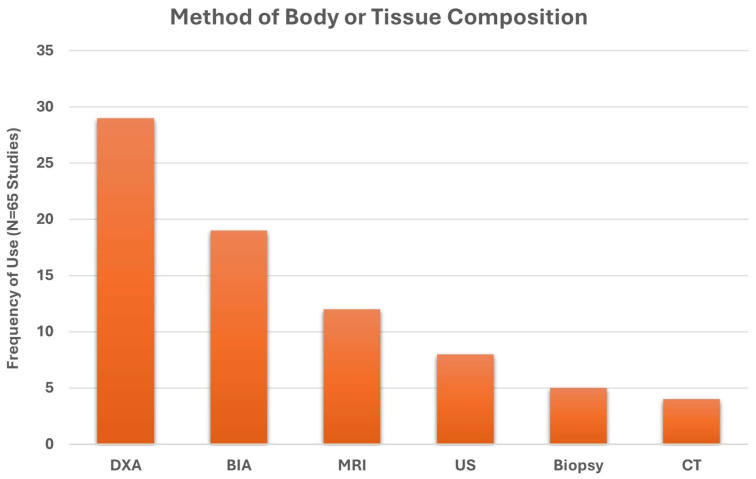
Identified methods (65 studies) of body and tissue composition assessment. DXA: dual-energy X-ray absorptiometry; BIA: bio-electrical impedance; MRI: magnetic resonance imaging; US: ultrasound; CT: computed tomography.

**Figure 8 jfmk-10-00367-f008:**
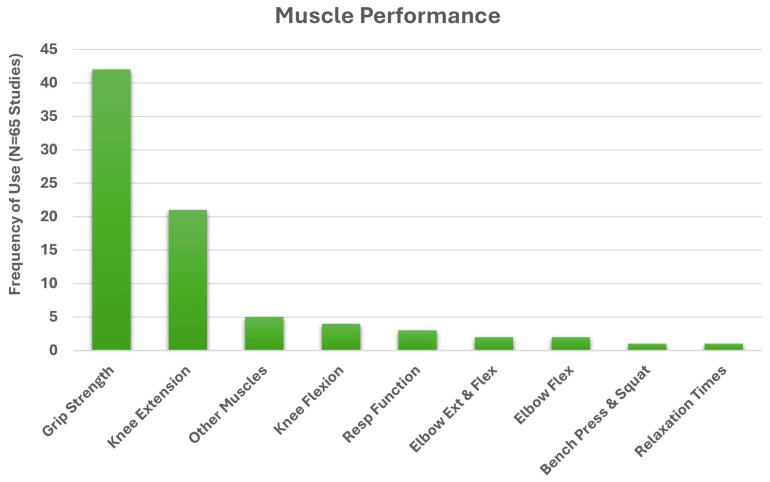
Identified methods (65 studies) of muscle performance assessment. Resp: respiratory; Flex: flexion.

**Figure 9 jfmk-10-00367-f009:**
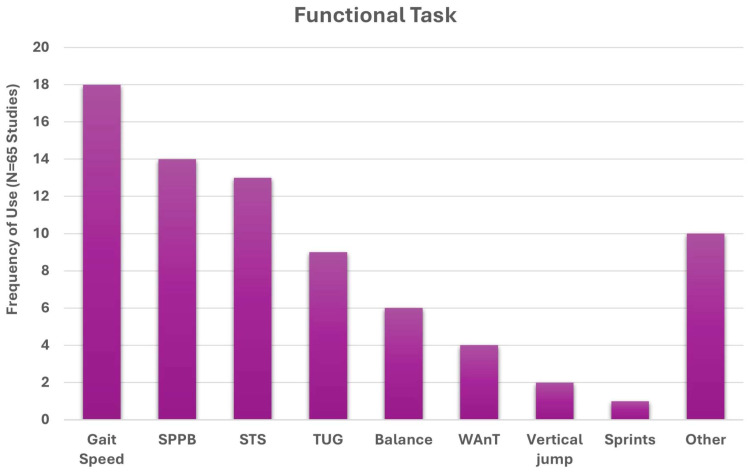
Identified methods (65 studies) of assessing ‘functional’ performance. SPPB: short physical performance battery; STS: sit-to-stand; TUG: timed up-and-go; WAnT: Wingate anaerobic test. See Table 1 for detailed ‘Other’ tests.

**Figure 10 jfmk-10-00367-f010:**
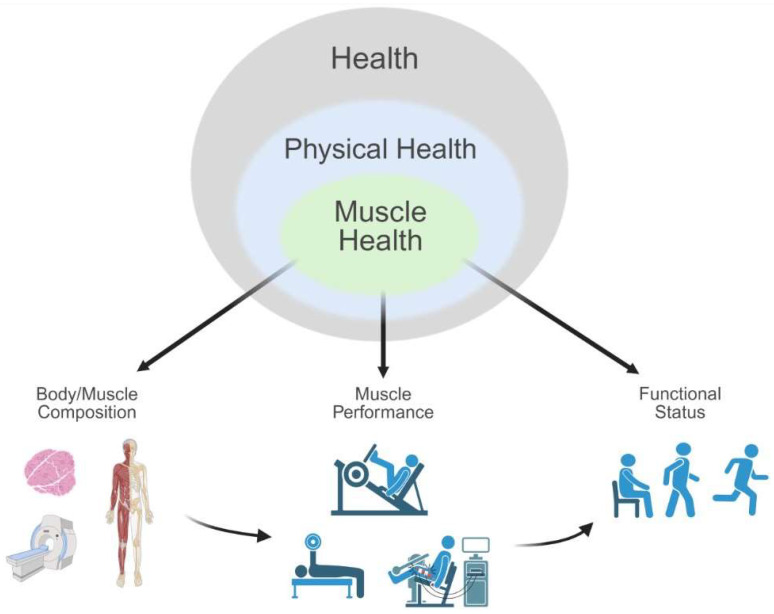
The proposed conceptual model of ‘muscle health’ as informed by the framework of the International Classification of Functioning, Disability and Health (ICF). Muscle health encompasses three primary domains: body/muscle tissue composition, muscular performance, and functional status. Domains can be evaluated using dichotomous (e.g., impaired vs. unimpaired; cut-off scores for functional assessments) or continuous metrics (e.g., maximal peak torque or force) depending on context and modality. This conceptual model emphasizes the integration of structural, physiological, and functional components relevant to muscle-related outcomes.

## Data Availability

Underlying data used to create figures are available upon request to the corresponding authors.

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
