# Peer review of "What Is ‘Muscle Health’? A Narrative Review and Conceptual Framework"

_jfmk, 2025, doi:10.3390/jfmk10040367_

Round 1
Reviewer 1 Report
Comments and Suggestions for Authors
Thank you to the authors for the opportunity to review this manuscript. The topic is highly relevant; however, there are significant methodological weaknesses that should be addressed.
I respectfully suggest the following considerations:
-
Title and Scope: The title should be reconsidered. It is important to clarify that the primary focus of this review is on the geriatric population, rather than the general population or individuals with athletic or musculoskeletal conditions.
-
Introduction: The introduction is overly long and, at times, redundant. The distinction between muscle health and muscle quality is not clearly articulated. It is important to recall that muscle quality is an umbrella term encompassing both contractile properties and muscular function. This includes the Muscle Quality Index (MQI), which serves as a functional or performance-related measure (Fragala et al., Muscle Quality in Aging: a Multi-Dimensional Approach to Muscle Functioning with Applications for Treatment, Sports Med 45, 641–658, 2015). In this regard, it remains unclear whether muscle quality is a subset of muscle health or if they are being used synonymously. The integration of the ICF framework into the discussion of muscle quality/health adds valuable clinical insight.
-
Methods: This is a critical area of concern. A systematic review should adhere to the PRISMA guidelines to ensure methodological transparency and replicability. Unfortunately, this review lacks key methodological details. It appears to rely solely on the keyword “muscle health” and restricts inclusion to studies published within the last five years, which limits comprehensiveness. The eligibility criteria appear subjective, and there is no assessment of the quality of the included evidence. Despite the evident effort invested in this manuscript, it reads more like a narrative literature review rather than a true systematic review. The absence of study registration (e.g., PROSPERO) and non-compliance with PRISMA further compromise its scientific utility.
Author Response
We thank the reviewers for their time and effort in performing a detailed review in ~1 week! We have addressed each comment and believe that the manuscript has been improved because of it. We hope the reviewer agrees! Please see each comment, and our replies below. Our replies are between ‘***’. Also, any new text in the manuscript is typed in red text.
Reviewer 1:
Title and Scope: The title should be reconsidered. It is important to clarify that the primary focus of this review is on the geriatric population, rather than the general population or individuals with athletic or musculoskeletal conditions.
***
We respectfully disagree with the reviewer for a few reasons. Firstly, the reviewer has picked up on the fact that a good number of the included reviews are in older populations. However, the age ranges of the study participants is quite large, and many studies include people with neurological, orthopedic, or other disorders or illnesses.
To make this clearer, we have added a demographic focused paragraph to the beginning of the Results section that reads (lines 282-298):
“A total of 16249 participants were included (n=7534 males, n=8628 females), with 55 studies including both males and females [31–34,36,39,40,42–44,46–48,50–68,70–72,74–86,88,89,91–95], and four [35,41,45,69], and six [37,38,49,73,87,90] studies being exclusively male and female, respectively. The included studies investigated a wide range of populations, spanning children as young as four years old [45] to adults in their 90s [48,79,92], with an average reported age of 61.6±16 years. However, the majority (n=43, 61.5%) of studies recruited older participants (≥60 years) [32–34,37–40,42,43,47,48,51–57,59,62,64–67,72,73,75,78–81,83–87,89–95], while the participants of 18 studies (27.7%) ranged from 36-59 years [31,35,36,41,44,46,49,50,58,60,61,63,69–71,74,76,82], and another four studies (6.1%) had participants with a mean age ≤35 years [45,68,77,88]. Most studies (n=33, 50.8%) focused on apparently healthy or community-dwelling individuals [32–35,37,39,41,44,47,49–51,53,55,57–59,61,64,66,67,72,73,75,77,80,81,86,90–94], while clinical cohorts (n=24, 37%) encompassed diverse conditions including cancer, COPD, CKD, musculoskeletal and neurological disorders, or chronic illness [31,38,40,42,43,45,48,52,54,56,62,63,65,68–71,74,76,78,84,85,89,95]. Only one study examined athletic (golfers) participants [82]. Seven studies (10.8%) included mixed or unclear populations [36,46,60,79,83,87,88].”
While we believe this new paragraph clarifies the populations nicely. We also think that the demographics involved are not particularly important as our article primarily aimed at summarizing the language around ‘muscle health’.
***
Introduction: The introduction is overly long and, at times, redundant. The distinction between muscle health and muscle quality is not clearly articulated. It is important to recall that muscle quality is an umbrella term encompassing both contractile properties and muscular function. This includes the Muscle Quality Index (MQI), which serves as a functional or performance-related measure (Fragala et al., Muscle Quality in Aging: a Multi-Dimensional Approach to Muscle Functioning with Applications for Treatment, Sports Med 45, 641–658, 2015). In this regard, it remains unclear whether muscle quality is a subset of muscle health or if they are being used synonymously. The integration of the ICF framework into the discussion of muscle quality/health adds valuable clinical insight.
***
We agree with this reviewer that the introduction was too long and potentially redundant in places. We have therefore removed 421 words (mostly from the last two paragraphs). Thus, even with the new text (see red text), the introduction is still ~175 words shorter than the previous version.
While we do not address the muscle quality comment in the introduction, we do briefly discuss in the discussion by writing:
“Similarly, terms like muscle ‘quality’ can have several meanings (e.g., composition, strength per unit of volume) which should be explored further.” (lines 623-625)
And
“Given the search criteria employed in this work, comparing “muscle health” to related concepts such as “muscle quality” was beyond the scope of this narrative review.” (lines 644-646)
***
Methods: This is a critical area of concern. A systematic review should adhere to the PRISMA guidelines to ensure methodological transparency and replicability. Unfortunately, this review lacks key methodological details. It appears to rely solely on the keyword “muscle health” and restricts inclusion to studies published within the last five years, which limits comprehensiveness. The eligibility criteria appear subjective, and there is no assessment of the quality of the included evidence. Despite the evident effort invested in this manuscript, it reads more like a narrative literature review rather than a true systematic review. The absence of study registration (e.g., PROSPERO) and non-compliance with PRISMA further compromise its scientific utility.
***
Due to the comments of this, and another reviewer, we have decided to alter the title (and review as a whole) to reflect the more ‘narrative’ nature of our review. This is reflected in the title, and several points in the abstract, methods, and discussion (esp. the limitations paragraph).
We hope that this change in title and tone will satisfy the reviewer’s concerns regarding the lack of inclusion of seminal works, risk-of-bias assessment, and pre-registration (note that the study was retrospectively registered with INPLASY)
***
Reviewer 2 Report
Comments and Suggestions for Authors
Please see the attached file.

Author Response
Reviewer 2:
I congratulate the authors and co-authors for their work. It is presented, with clear and concise aims and definitions, the wording and structure is of very good quality, and the topic is of relevance as outlined throughout the manuscript. Therefore, I only have some minor comments which I hope helps to improve the quality of the manuscript.
***
We thank the reviewer for quickly performing a detailed review of our paper. We are happy that the reviewer sees value in our work, and we hope that we have fully addressed each of their critiques.
***
Lines 31-36: the sentence seem to be incomplete, please revise.
***
We are thankful that the reviewer has identified a poorly worded sentence (esp. in the most visible part of the paper).
We have re-worded these sentences, which now read (lines 34-37):
“Within the framework of the International Classification of Functioning, Disability and Health, these categories aligned with muscle health components of muscle morphology/morphometry (e.g., mass and composition), functional status (performance-based tasks), and physical capacity (muscle performance).”
***
Lines 61-63: I would also suggest including strength performance as an outcome measure for the general population.
***
We thank the reviewer for the helpful comment. We now write (lines 82-84):
“Furthermore, minimally time-consuming muscular performance tests (e.g., grip strength) may be warranted in the general population to serve as a proxy for longevity, quality of life, and cellular health [10–12].”
While citing:
Peterson, M.D.; Collins, S.; Meier, H.C.S.; Brahmsteadt, A.; Faul, J.D. Grip Strength Is Inversely Associated with DNA Methylation Age Acceleration. J cachexia sarcopenia muscle 2023, 14, 108–115, doi:10.1002/jcsm.13110.
Chai, L.; Zhang, D.; Fan, J. Comparison of Grip Strength Measurements for Predicting All-Cause Mortality among Adults Aged 20+ Years from the NHANES 2011–2014. Sci Rep 2024, 14, 29245, doi:10.1038/s41598-024-80487-y.
Kaczorowska, A.; Kozieł, S.; Ignasiak, Z. Hand Grip Strength and Quality of Life among Adults Aged 50–90 Years from South West Poland. Sci Rep 2025, 15, 882, doi:10.1038/s41598-024-84923-x.
Lines 103-104: a brief description of what are the components of both ADL and IADL are needed.
***
The reviewer is correct that brief descriptions of ADL and IADL are important, especially for relatively lay readers. Thus, we now write (lines 122-127:
“Multiple investigators have observed that declines in muscle strength are frequently associated with diminished performance in activities of daily living (ADL), such as bathing, dressing, eating, toileting, and ambulation, and instrumental activities of daily living (IADL), which include more complex tasks including managing finances, shopping, meal preparation, housekeeping, and medication management [16,17].”
***
Line 120: what do authors consider low muscle mass levels?
***
We now write (lines 141-143):
“...typically defined using appendicular lean mass indexed to height squared, with Europe-an Working Group on Sarcopenia in Older People (EWGSOP) recommended cut-points of <7.0 kg/m² in men and <5.5 kg/m² in women [20]…”
***
Lines 198-200: 'muscle health' appears three times in the same sentence, please revise and reword to avoid being repetitive.
***
We thank the reviewer for the attention to detail. We have re-written this section to avoid repetition.
The section now reads (lines 189-194):
“A narrative review was conducted by identifying papers to better understand the conceptual and operational definition of ‘muscle health’ used by other investigators and document the assessment tools used. These study data and definitional terms were then extracted and combined where appropriate to synthesize the current use of the term ‘muscle health’. This information was then interpreted using the ICF framework to develop a proposed conceptual model.”
***
Lines 207-209: were athletic population studies included or excluded?
***
We now clarify by writing (lines 205-206):
“Otherwise, the conceptual nature of this review lead us to include any randomized controlled or clinical trial, regardless of demographics (e.g., age, disease-state, athletes).”
We have also added a paragraph to the beginning of the results section that breaks down the study demographics. This new paragraph reads (lines 282-298):
“A total of 16249 participants were included (n=7534 males, n=8628 females), with 55 studies including both males and females [31–34,36,39,40,42–44,46–48,50–68,70–72,74–86,88,89,91–95], and four [35,41,45,69], and six [37,38,49,73,87,90] studies being exclusively male and female, respectively. The included studies investigated a wide range of populations, spanning children as young as four years old [45] to adults in their 90s [48,79,92], with an average reported age of 61.6±16 years. However, the majority (n=43, 61.5%) of studies recruited older participants (≥60 years) [32–34,37–40,42,43,47,48,51–57,59,62,64–67,72,73,75,78–81,83–87,89–95], while the participants of 18 studies (27.7%) ranged from 36-59 years [31,35,36,41,44,46,49,50,58,60,61,63,69–71,74,76,82], and another four studies (6.1%) had participants with a mean age ≤35 years [45,68,77,88]. Most studies (n=33, 50.8%) focused on apparently healthy or community-dwelling individuals [32–35,37,39,41,44,47,49–51,53,55,57–59,61,64,66,67,72,73,75,77,80,81,86,90–94], while clinical cohorts (n=24, 37%) encompassed diverse conditions including cancer, COPD, CKD, musculoskeletal and neurological disorders, or chronic illness [31,38,40,42,43,45,48,52,54,56,62,63,65,68–71,74,76,78,84,85,89,95]. Only one study examined athletic (golfers) participants [82]. Seven studies (10.8%) included mixed or unclear populations [36,46,60,79,83,87,88].”
***
Lines 348: the y axis need a measurement unit, also a would recommend a figure of better quality (it just seem to be extracted from excel) … Same for figure 6, 7, 8.
***
We thank the reviewer for their attention to detail. We have added a Y-axis to each bar graph that reads “Frequency of use (N=65 studies)“ (the figure captions have also been edited to include this detail.
The reviewer is also correct that the figures were originally extracted straight from excel. Thus, we have now exported from excel to PowerPoint and saved each figure as 300 DPI. We hope that the reviewer is now satisfied with the upgraded quality of our figures.
***
Line 425: TUG must be defined prior to its first use.
***
We have identified the first use of ‘TUG’ in the text (not figures or tables) as the 4th paragraph of the results and have now defined (line 351).
***
Lines 451-454: authors must acknowledge that no precise measurements of lean or muscle exists to date, even DXAs have several limitations. Skinfolds, BIA, all of them have pros and cons, but none is precise.
***
We have addressed the reviewers correct comment by adding two sentences that read (lines 484-488):
“Nevertheless, no current non-invasive method offers an exact quantification of skeletal muscle mass, with each approach, including DXA, BIA, skinfolds, CT, MRI, ultra-sound, and even emerging techniques such as D3-creatine, carrying inherent assumptions and limitations [105]. Accordingly, these methods should be viewed as providing useful, but imperfect proxies, each with unique strengths and drawbacks.”
We cite:
Buckinx, F.; Landi, F.; Cesari, M.; Fielding, R.A.; Visser, M.; Engelke, K.; Maggi, S.; Dennison, E.; Al‐Daghri, N.M.; Allepaerts, S.; et al. Pitfalls in the Measurement of Muscle Mass: A Need for a Reference Standard. J Cachexia Sarcopenia Muscle 2018, 9, 269–278, doi:10.1002/jcsm.12268.
***
Line 490: the use of 'functional' is confusing in this context. What is considered functional? or examples of such tasks may help to clarify this for the general reader.
***
We thank the reviewer for pointing this out. We have revised the paragraph to clarify what we mean by “functional” in this context, specifying that it refers to mobility-related tasks relevant to activities of daily living, such as walking, chair rises, or stair climbing. Examples have been added to make this section clearer for the general reader.
We write:
“These methods provide meaningful information on lower limb strength, balance, and overall mobility, which can directly impact ADL. By combining selected functional tasks through assessment batteries, such as the SPPB, one can obtain a comprehensive assessment of functional status, reflecting an individual’s ability to perform these mobility-related activities. Nevertheless, the multi-system contributions to functional status require an appropriate patient history and physical exam to determine if muscle dysfunction is a key contributor to observed functional limitations and diminished mobility. Additionally, functional tests vary in their relative difficulty and bias towards either muscle strength or power. For example, tasks with a focus on muscle power, such as the 30-second chair rise test, may reveal performance deficits earlier than less demanding tasks, such as usual gait speed [116].”
***
Reviewer 3 Report
Comments and Suggestions for Authors
I hope this letter finds you well. I had the opportunity to review your article titled, “What is ‘muscle health’? A systematic review and conceptual framework.
- Introduction
- It is well-known that the concept of “muscle health” is frequently used in clinical and research settings, but there is no consistent definition.
- The necessity of this study is persuasively presented by emphasizing that the conceptual confusion in this study negatively affects clinical application and interpretation of research results.
- Describes the functional importance of muscles in connection with various areas such as metabolism, recovery, and geriatric health.
- It is original to propose that muscle function indicators can be proposed as a new “vital sign” in comparison to traditional geriatric medical examination items (blood pressure, pulse, etc.).
- By presenting the WHO's ICF (International Classification of Functioning, Disability and Health) as a framework of evidence, we have secured academic legitimacy in presenting a conceptual framework that goes beyond simple terminological definitions.
- However, although the difference from existing concepts related to ‘muscle health’ (e.g., muscle quality, sarcopenia, frailty, etc.) was mentioned, the relationship between these terms was not compared or criticized in more depth.
- It suggests a connection between ICF and muscle health, but the logical basis for the specific “why ICF is the most appropriate framework” is weak.
- Therefore, we propose improvements as follows.
- It can make the argument “why a new concept definition is needed” more persuasive.
- You can be more persuasive to readers and reviewers by presenting specific data, such as the increasing frequency of use of the term “muscle health” over the past five years and cases of inconsistency in RCTs.
- It would be good to clarify the reasons for choosing the ICF and highlight its advantages by comparing it with other possible frameworks (e.g., Nagi model, disablement process model, etc.).
- Method
- The authors of this study clearly stated that they conducted their study according to the Preferred Reporting Items for Systematic Reviews and Meta-Analyses (PRISMA) guidelines and presented a PRISMA flow diagram (Figure 2). This is considered an important element in ensuring transparency in systematic reviews.
- The authors used two major databases, PubMed and CINAHL, and the search keyword was clearly defined as “muscle health,” which increases reproducibility.
- The search period was limited to the last five years (last search: March 2025) to ensure the most up-to-date literature was reflected.
- The reliability of the study is enhanced by the fact that three independent reviewers reviewed the title/abstract and text, and third-party consensus was reached in case of disagreement.
- Systematic filtering was performed by specifying inclusion criteria (clinical and RCTs, human subjects, including muscle health measurements) and exclusion criteria (non-human, case studies, review articles, lack of results).
- Describe the process of literature management and collaboration using tools such as Zotero and Covidence.
- The provision of definitions, measurement tools, and key results during data extraction were specifically organized, and the fact that the code was coded based on Excel increases the possibility of reproducibility.
- Through a mixed qualitative and quantitative analysis (frequency analysis + thematic content analysis), we derived a conceptual model that went beyond a simple summary.
- However, since only the single keyword “muscle health” was used and related concepts such as “muscle quality,” “sarcopenia,” and “muscle function” were not included, there is a possibility that related studies may have been omitted.
- The authors explained that it was “unnecessary given the nature of a conceptual review,” but given that it takes the form of a systematic review, it is desirable to mention whether a minimal bias assessment tool (e.g., Cochrane risk-of-bias tool, ROBINS-I) was applied.
- Although “frequency, co-occurrence, contextual integration” is mentioned, it is unclear whether this is a systematic qualitative analysis (e.g., thematic synthesis, framework analysis).
- Results
- This study systematically classified the definition, measurement items, and measurement tools of ‘muscle health’ among 68 studies.
- Organizing the results into four main categories (① muscle mass/body composition, ② muscle performance, ③ physical function, ④ tissue composition) increased understandability and comparability.
- This objectively presents the current status by presenting the percentage of frequency with which various measurement methods, such as DXA, grip strength, gait speed, MRI, and CT, are used. This is useful for identifying which indicators are most widely used in clinical and research settings.
- It was pointed out that only 31 of the 68 articles provided an ‘operational definition’, and the remaining 37 articles only performed measurements without a definition.
- The authors presented a PRISMA flow chart, word cloud, and frequency graph (e.g., Figure 3–8) to intuitively aid data interpretation.
- However, in-depth analysis/critical discussions such as “Why was body composition used the most?” and “Why was the definition lacking?” were not sufficiently covered in the results section.
- It would have been more convincing if there had been ‘subgroup analysis’ rather than simple overall frequency.
- For the 37 studies that only measured without defining the results, it would have been more persuasive if the limitations and common trends had been evaluated more deeply.
- Discussion
- It clearly points out the lack of consistent definitions in the existing literature and emphasizes the need for a proposed conceptual framework to address this.
- It is judged to have high academic value in that muscle health is not viewed simply as muscle mass and function, but is reconstructed into a multidimensional framework of morphology/morphometry, functional status, and physical capacity.
- The authors discussed that once a standardized definition and measurement indicators for muscle health are established, it can contribute to various clinical areas such as geriatric health care, chronic disease management, and rehabilitation medicine.
- It is positive that it goes beyond simply organizing academic concepts and mentions the possibility of clinical application.
- By mentioning the possibility of linking with adjacent concepts such as resilience, intrinsic capacity, and muscle quality, it suggests that the muscle health framework can complement and integrate existing health concepts.
- However, we reiterate that the definition and measurement of the concept of muscle health are inconsistent, and there is a lack of in-depth root cause analysis of why this inconsistency has occurred.
- While the need for standardization is emphasized, there is a lack of a concrete roadmap for which indicators should be designated as “core indicators” and how they should be applied in clinical settings.
- The limitations of the search scope (only PubMed and CINAHL were used), the simplicity of the search terms (a single keyword of ‘muscle health’), and the failure to conduct a risk of bias assessment mentioned in the Methods section were not sufficiently reflected in the Results and Discussion sections.
Author Response
Reviewer 3:
I hope this letter finds you well. I had the opportunity to review your article titled, “What is ‘muscle health’? A systematic review and conceptual framework.
***
We thank the reviewer for their timely turnaround with detailed comments and critiques of our article. We hope that the reviewer is fully satisfied with our revisions.
***
Introduction
- It is well-known that the concept of “muscle health” is frequently used in clinical and research settings, but there is no consistent definition.
- The necessity of this study is persuasively presented by emphasizing that the conceptual confusion in this study negatively affects clinical application and interpretation of research results.
- Describes the functional importance of muscles in connection with various areas such as metabolism, recovery, and geriatric health.
- It is original to propose that muscle function indicators can be proposed as a new “vital sign” in comparison to traditional geriatric medical examination items (blood pressure, pulse, etc.).
- By presenting the WHO's ICF (International Classification of Functioning, Disability and Health) as a framework of evidence, we have secured academic legitimacy in presenting a conceptual framework that goes beyond simple terminological definitions.
- However, although the difference from existing concepts related to ‘muscle health’ (e.g., muscle quality, sarcopenia, frailty, etc.) was mentioned, the relationship between these terms was not compared or criticized in more depth.
- It suggests a connection between ICF and muscle health, but the logical basis for the specific “why ICF is the most appropriate framework” is weak.
***
We thank the reviewer for (mostly positively) summarizing the Introduction section. The reviewer provides three constructive points for improvement. Please see each point below, with our reply.
***
- Therefore, we propose improvements as follows.
- It can make the argument “why a new concept definition is needed” more persuasive.
***
In line with the reviewers next comment, we highlight the growth of ‘muscle health’ by writing (lines 49-56):
“For example, PubMed records show that use of the term ‘muscle health’ has increased substantially over the past two decades, with more than a fourfold rise in publications between 2010–2014 (5709±1411 hits) and 2020–2024 (16040±1130 hits) (Figure 1), with >14,000 hits through September 2025 alone, highlighting its growing prominence. However, while increasing at a substantial rate (like ‘reproductive health’, ‘joint health’, and ‘bone health’, but behind ‘cardiovascular health’ [see Figure 1]), muscle health remains inconsistently defined compared to more established constructs.”
Please note the new figure.
We also now finish this same paragraph by writing (lines 65-72):
“This lack of clarity is particularly concerning given the rapid rise in publications using the term muscle health (see Figure 1). As with more established constructs such as bone health and cardiovascular health, the growing use of the term without a standardized framework risks diluting its meaning, creating confusion in both clinical and research contexts. Establishing a definition and conceptual model of muscle health at this stage is therefore critical to ensure that its growing prominence is matched by scientific rigor and clinical utility.”
***
- You can be more persuasive to readers and reviewers by presenting specific data, such as the increasing frequency of use of the term “muscle health” over the past five years and cases of inconsistency in RCTs.
***
“We thank the reviewer for the helpful suggestion. We now report the number and rise in PubMed hits for ‘muscle health’ (and, for context, other health types).
We now write (lines 49-56):
“For example, PubMed records show that use of the term ‘muscle health’ has increased substantially over the past two decades, with more than a fourfold rise in publications between 2010–2014 (5709±1411 hits) and 2020–2024 (16040±1130 hits) (Figure 1), with >14,000 hits through September 2025 alone, highlighting its growing prominence. How-ever, while increasing at a substantial rate (like ‘reproductive health’, ‘joint health’, and ‘bone health’, but behind ‘cardiovascular health’ [see Figure 1]), muscle health remains inconsistently defined compared to more established constructs.”
***
- It would be good to clarify the reasons for choosing the ICF and highlight its advantages by comparing it with other possible frameworks (e.g., Nagi model, disablement process model, etc.).
***
We have addressed the reviewers comment by adding a few sentences to the end of the first paragraph under the 1.2. subsection.
These sentences read (lines 170-179):
“The ICF framework was selected because of its broad adoption in clinical and rehabilitation contexts, international recognition, biopsychosocial framework, and provision of a standardized taxonomy that facilitates comparison across populations and health conditions. In contrast, earlier conceptual models, such as the Nagi model, which primarily emphasized progression from pathology to disability [29], did not offer the same level of operational detail for the non-linear integration of body-level impairments, tissue characteristics, and activity limitations. Additionally, the ICF is a progression from the original International Classification of Impairments, Disabilities, and Handicaps model and has greater adoption than other disablement models such as the Institute of Medicine Enablement-Disablement Process Model. ”
We cite:
Forstner, M. Conceptual Models of Disability: The Development of the Consideration of Non-Biomedical Aspects. Disabilities 2022, 2, 540–563, doi:10.3390/disabilities2030039.
***
Method
- The authors of this study clearly stated that they conducted their study according to the Preferred Reporting Items for Systematic Reviews and Meta-Analyses (PRISMA) guidelines and presented a PRISMA flow diagram (Figure 2). This is considered an important element in ensuring transparency in systematic reviews.
- The authors used two major databases, PubMed and CINAHL, and the search keyword was clearly defined as “muscle health,” which increases reproducibility.
- The search period was limited to the last five years (last search: March 2025) to ensure the most up-to-date literature was reflected.
- The reliability of the study is enhanced by the fact that three independent reviewers reviewed the title/abstract and text, and third-party consensus was reached in case of disagreement.
- Systematic filtering was performed by specifying inclusion criteria (clinical and RCTs, human subjects, including muscle health measurements) and exclusion criteria (non-human, case studies, review articles, lack of results).
- Describe the process of literature management and collaboration using tools such as Zotero and Covidence.
- The provision of definitions, measurement tools, and key results during data extraction were specifically organized, and the fact that the code was coded based on Excel increases the possibility of reproducibility.
- Through a mixed qualitative and quantitative analysis (frequency analysis + thematic content analysis), we derived a conceptual model that went beyond a simple summary.
***
We again thank the reviewer for a positive summary of the Methods section. We are especially happy that the reviewer has pointed out the strength of our derived conceptual model beyond only reporting summary statistics.
The reviewer does have some points of critique which we address in detail below.
***
Thank you for this insightful comment. We agree that related concepts such as “muscle quality,” “sarcopenia,” and “muscle function” are highly relevant to the broader field of muscle research. However, the explicit aim of our review was to investigate how the term “muscle health” is currently defined and operationalized in the literature. For this reason, we intentionally restricted our search strategy to studies that used “muscle health” as a keyword or descriptor. This focus allowed us to assess the construct validity and consistency of this emerging terminology, rather than to synthesize all muscle-related constructs, which have been the focus of multiple prior reviews. We have clarified this rationale in a few places throughout the manuscript.
Abstract:
We have added (lines 21-22):
“...explicitly described under the term ‘muscle health’.”
Methods (lines 222-224):
“We intentionally limited our search to studies that explicitly used the term ‘muscle health’, as the primary aim of this review was to examine how this emerging terminology is being defined and operationalized.”
Discussion (lines 641-646):
“Second, by restricting our search to studies that explicitly used the term “muscle health”, we may have excluded research employing closely related constructs (e.g., “muscle quality,” “sarcopenia,” or “muscle function”); however, this was a deliberate methodological decision to examine how the specific term “muscle health” is currently defined and operationalized.”
***
- The authors explained that it was “unnecessary given the nature of a conceptual review,” but given that it takes the form of a systematic review, it is desirable to mention whether a minimal bias assessment tool (e.g., Cochrane risk-of-bias tool, ROBINS-I) was applied.
***
Due to the comments from this, and the other 2 reviewers, we agree that this article is more accurately presented as a “narrative review”. We still believe that a risk-of-bias assessment is not relevant to the goal of the review, but are hopeful that this reviewer will agree based on the change of title and tone of the review to better reflect its narrative focus.
***
- Although “frequency, co-occurrence, contextual integration” is mentioned, it is unclear whether this is a systematic qualitative analysis (e.g., thematic synthesis, framework analysis).
***
We have altered this paragraph to more clearly discuss our analysis. The entire paragraph now reads (lines 244-263):
“We employed a mixed synthesis approach combining quantitative tallying of measurement domains with structured qualitative description to develop a proposed model of muscle health using the ICF framework. Each identified study was reviewed for the inclusion of assessments of body composition, tissue composition, muscle performance, and functional performance. Common language from operational definitions was extracted and analyzed in an Excel spreadsheet (see Figure 3). Absolute frequencies of inclusion across these domains were calculated (see Figures 4–8), with relative (percentages) re-ported in-text. These data were interpreted in conjunction with the ICF framework and prior theoretical models [27], enabling us to identify recurring elements and their contextual applications. While we did not conduct a formal qualitative synthesis (e.g., thematic or framework analysis), the frequency, co-occurrence, and descriptive integration of do-mains across studies informed the proposed components (body/muscle tissue composition, muscle performance, functional performance) for the model. The development of the conceptual muscle health model employed a flexible approach that provides proposed domains and component categories suitable for both clinical and research applications. Nevertheless, the final selection of tests and measures used to assess muscle health components and the recommended data interpretation standards are beyond the scope of this work. The identification of assessment standards consistent with a conceptual model of muscle health is subject to further research and future consensus efforts. A risk-of-bias analysis was unnecessary due to the conceptual/narrative nature of the present review.”
Please see the manuscript file to see the red text for specific edits.
***
Results
- This study systematically classified the definition, measurement items, and measurement tools of ‘muscle health’ among 68 studies.
- Organizing the results into four main categories (① muscle mass/body composition, ② muscle performance, ③ physical function, ④ tissue composition) increased understandability and comparability.
- This objectively presents the current status by presenting the percentage of frequency with which various measurement methods, such as DXA, grip strength, gait speed, MRI, and CT, are used. This is useful for identifying which indicators are most widely used in clinical and research settings.
- It was pointed out that only 31 of the 68 articles provided an ‘operational definition’, and the remaining 37 articles only performed measurements without a definition.
- The authors presented a PRISMA flow chart, word cloud, and frequency graph (e.g., Figure 3–8) to intuitively aid data interpretation.
***
We again thank the reviewer for the complementary summary of the section.
Please see the specific critiques, with our replies below.
***
- It would have been more convincing if there had been ‘subgroup analysis’ rather than simple overall frequency.
***
We have also added a new paragraph that covers the demographics of the included studies. We write (lines 282-298):
“A total of 16249 participants were included (n=7534 males, n=8628 females), with 55 studies including both males and females [31–34,36,39,40,42–44,46–48,50–68,70–72,74–86,88,89,91–95], and four [35,41,45,69], and six [37,38,49,73,87,90] studies being exclusively male and female, respectively. The included studies investigated a wide range of populations, spanning children as young as four years old [45] to adults in their 90s [48,79,92], with an average reported age of 61.6±16 years. However, the majority (n=43, 61.5%) of studies recruited older participants (≥60 years) [32–34,37–40,42,43,47,48,51–57,59,62,64–67,72,73,75,78–81,83–87,89–95], while the participants of 18 studies (27.7%) ranged from 36-59 years [31,35,36,41,44,46,49,50,58,60,61,63,69–71,74,76,82], and another four studies (6.1%) had participants with a mean age ≤35 years [45,68,77,88]. Most studies (n=33, 50.8%) focused on apparently healthy or community-dwelling individuals [32–35,37,39,41,44,47,49–51,53,55,57–59,61,64,66,67,72,73,75,77,80,81,86,90–94], while clinical cohorts (n=24, 37%) encompassed diverse conditions including cancer, COPD, CKD, musculoskeletal and neurological disorders, or chronic illness [31,38,40,42,43,45,48,52,54,56,62,63,65,68–71,74,76,78,84,85,89,95]. Only one study examined athletic (golfers) participants [82]. Seven studies (10.8%) included mixed or unclear populations [36,46,60,79,83,87,88].”
We have also added several sentences to better describe the 36 studies that did not provide a definition, so that readers can compare the rate of assessment types between these study groups.
We write (lines 338-345):
“The 36 studies that used, but did not define ‘muscle health’, had a similar assessment distribution to the studies that provided an operational definition. Most studies have emphasized body composition or muscle mass (N=31, 86.1%) [32–34,37,38,40,42,44,47,48,51,52,54,57,58,66–68,72,74,77,79–82,84,85,92–95], and muscle performance (N=31, 86.1%) [32–34,37,38,40,42,44,46–48,51,52,54,57,58,66,67,72,74,77,79–82,85,90–93,95], with fewer incorporating functional tasks (N=23, 63.9%) [33,34,37,38,40,42,44,47,51,52,57,60,66–68,72,79,82,84,85,91,93,95], or tissue composition (N=7, 19.4%) [32,34,44,46,57,58,90].”
***
- For the 36 studies that only measured without defining the results, it would have been more persuasive if the limitations and common trends had been evaluated more deeply.
***
We have now added a set of sentences that summarizes the tests used in those 36 studies. It reads (lines 338-345):
“The 36 studies that used, but did not define ‘muscle health’, had a similar assessment distribution to the studies that provided an operational definition. Most studies have emphasized body composition or muscle mass (N=31, 86.1%) [40–42,45,46,48,50,52,55,56,59,60,62,65,66,74–76,80,82,85,87–90,92,93,100–103], and muscle performance (N=31, 86.1%) [40–42,45,46,48,50,52,54–56,59,60,62,65,66,74,75,80,82,85,87–90,93,98–101,103], with fewer incorporating functional tasks (N=23, 63.9%) [41,42,45,46,48,50,52,55,59,60,65,68,74–76,80,87,90,92,93,99,101,103], or tissue composition (N=7, 19.4%) [40,42,52,54,65,66,98].”
We have also added a few sentences to the discussion that points out the pitfalls/limitations of these studies and why measures should be defined.
These sentences read (lines 434-444):
“A key limitation across the studies that used, but did not define ‘muscle health’ is the lack of conceptual clarity. ‘Muscle health” was often used interchangeably with sarcopenia, muscle quality, or general musculoskeletal status, without justification for why particular measures were included or excluded. This inconsistency undermines comparability across studies, as similar outcomes were variably treated as either central or peripheral to “muscle health.” Furthermore, reliance on surrogate markers such as BMI or broad functional questionnaires, without integration into a clear conceptual framework, makes it difficult to interpret whether observed associations truly reflect skeletal muscle status. These limitations underscore the importance of establishing a standardized framework, so that “muscle health” assessments can be applied consistently in both clinical and research contexts.”
***
Discussion
- It clearly points out the lack of consistent definitions in the existing literature and emphasizes the need for a proposed conceptual framework to address this.
- It is judged to have high academic value in that muscle health is not viewed simply as muscle mass and function, but is reconstructed into a multidimensional framework of morphology/morphometry, functional status, and physical capacity.
- The authors discussed that once a standardized definition and measurement indicators for muscle health are established, it can contribute to various clinical areas such as geriatric health care, chronic disease management, and rehabilitation medicine.
- It is positive that it goes beyond simply organizing academic concepts and mentions the possibility of clinical application.
- By mentioning the possibility of linking with adjacent concepts such as resilience, intrinsic capacity, and muscle quality, it suggests that the muscle health framework can complement and integrate existing health concepts.
***
We are grateful for the reviewers comments throughout, and are happy that they see the value in our work, including the discussion.
We address the specific criticisms for the discussion section below.
***
- However, we reiterate that the definition and measurement of the concept of muscle health are inconsistent, and there is a lack of in-depth root cause analysis of why this inconsistency has occurred.
- While the need for standardization is emphasized, there is a lack of a concrete roadmap for which indicators should be designated as “core indicators” and how they should be applied in clinical settings.
***
This is another great comment from the reviewer which helped to improve practical application from this paper.
We have added a new paragraph under a new subsection and write (lines: 607-637)
“4.3. Toward core indicators of muscle health
While this review highlights substantial variability in definitions and measurement methods, several domains consistently emerged across studies and align with existing consensus recommendations in sarcopenia and physical function research. Based on frequency of use, psychometric strength, and feasibility, we propose that the identified domains of (1) Body Systems/Structures (i.e., body/muscle tissue composition and muscle performance) and (2) Participation (i.e., functional status) provide a foundation to develop assessment guidelines for clinical and research applications (see Figure 10). Assessment tools corresponding with these domains have well-documented associations with health outcomes, hold solid psychometric properties, and can be implemented across a range of settings, reflecting the multidimensionality of muscle health.
The use of simple standardized assessment tools in clinical settings does not preclude the adoption of more advanced measures in research settings (e.g., grip strength testing versus isokinetic dynamometry). While advanced assessment tools for muscle performance or tissue characteristics are appropriate in specialized contexts, accounting for the availability of both simple and advanced methods will facilitate the creation of a practical roadmap towards standardized assessment guidelines. Similarly, terms like muscle ‘quality’ can have several meanings (e.g., composition, strength per unit of volume) which should be explored further. In clinical settings, prioritizing feasible and validated assessment tools (e.g., grip strength, gait speed) ensures broad applicability. In research contexts, the scope may be expanded to include more detailed compositional and performance-based measures to aid mechanistic research. Importantly, a practical roadmap towards standardized assessment guidelines for muscle health includes addressing the issues of data acquisition and interpretation. This effort encompasses a range of tasks, from addressing data processing issues and developing specific protocols for performance-based tests to normalizing strength measurements based on body size or muscle volume. Gaining clarity on data acquisition and interpretation issues related to assessing muscle health will require further consensus efforts and additional methodological studies. Nonetheless, this tiered approach to standardized assessment methods across practice settings may improve comparability across studies while maintaining flexibility for both practitioners and investigators.”
***
- The limitations of the search scope (only PubMed and CINAHL were used), the simplicity of the search terms (a single keyword of ‘muscle health’), and the failure to conduct a risk of bias assessment mentioned in the Methods section were not sufficiently reflected in the Results and Discussion sections.
***
We previously mentioned and clarified why a risk of bias assessment was not done (including a change from labeling the review as ‘systematic’ to ‘narrative’. We hope that the reviewer is satisfied in this regard.
We have also added new points to the limitations (and methods: see above) to further address the reviewers’ points. The new limitations section now reads (see red text for specific additions):
“Despite the comprehensive nature of this review, several limitations must be acknowledged. First, many studies inferred definitions of muscle health through outcomes without explicitly defining the term. Second, by restricting our search to studies that explicitly used the term “muscle health”, we may have excluded research employing closely related constructs (e.g., “muscle quality,” “sarcopenia,” or “muscle function”); however, this was a deliberate methodological decision to examine how the specific term “muscle health” is currently defined and operationalized. Furthermore, reliance on specific databases (only CINAHL and PubMed) may have introduced bias in the selection of studies, potentially overlooking pertinent research published elsewhere. In addition, het erogeneity in study design and participant samples makes generalizing the findings across all demographic groups challenging. Most importantly, although we conducted a systematic search to assess the current literature, the overarching narrative format of this review is susceptible to bias due to the authors' perspective in the manuscript. Thus, our viewpoints are not infallible, and this paper is open to further and differing interpretations. Lastly, our review focused primarily on skeletal muscle health, which limits the generalizability of our findings to other muscle types, such as cardiac or smooth muscle.”
Thus, the concerns about search terms and databases have been addressed.
***
Round 2
Reviewer 2 Report
Comments and Suggestions for Authors
The authors have properly addressed all my comments and suggestions.